# Long Sequence Hopfield Memory

**Hamza Tahir Chaudhry**[1,2], **Jacob A. Zavatone-Veth**[2,3],
**Dmitry Krotov**[5], **Cengiz Pehlevan**[1,2,4]

[1]John A. Paulson School of Engineering and Applied Sciences,
[2]Center for Brain Science, [3]Department of Physics,
[4]Kempner Institute for the Study of Natural and Artificial Intelligence,
Harvard University
Cambridge, MA 02138
[5]MIT-IBM Watson AI Lab, IBM Research,
Cambridge, MA 02142
`hchaudhry@g.harvard.edu, jzavatoneveth@g.harvard.edu,`
`krotov@ibm.com, cpehlevan@seas.harvard.edu`

## Abstract

Sequence memory is an essential attribute of natural and artificial intelligence that enables agents to encode, store, and retrieve complex sequences of stimuli and actions. Computational models of sequence memory have been proposed where recurrent Hopfield-like neural networks are trained with temporally asymmetric Hebbian rules. However, these networks suffer from limited sequence capacity (maximal length of the stored sequence) due to interference between the memories. Inspired by recent work on Dense Associative Memories, we expand the sequence capacity of these models by introducing a nonlinear interaction term, enhancing separation between the patterns. We derive novel scaling laws for sequence capacity with respect to network size, significantly outperforming existing scaling laws for models based on traditional Hopfield networks, and verify these theoretical results with numerical simulation. Moreover, we introduce a generalized pseudoinverse rule to recall sequences of highly correlated patterns. Finally, we extend this model to store sequences with variable timing between states' transitions and describe a biologically-plausible implementation, with connections to motor neuroscience.

## 1   Introduction

Memory is an essential ability of intelligent agents that allows them to encode, store, and retrieve information and behaviors they have learned throughout their lives. In particular, the ability to recall sequences of memories is necessary for a large number of cognitive tasks with temporal or causal structure, including navigation, reasoning, and motor control [1–9].

Computational models with varying degrees of biological plausibility have been proposed for how neural networks can encode sequence memory [1–3, 10–22]. Many of these are based on the concept of associative memory, also known as content-addressable memory, which refers to the ability of a system to recall a set of objects or ideas when prompted by a distortion or subset of them. Modeling associative memory has been an extremely active area of research in computational neuroscience and deep learning for many years, with the Hopfield network becoming the canonical model [23–25].

Unfortunately, a major limitation of the traditional Hopfield Network and related associative memory models is its capacity: the number of memories it can store and reliably retrieve scales linearly with the number of neurons in the network. This limitation is due to interference between different memories during recall, also known as crosstalk, which decreases the signal-to-noise ratio. Large amounts of crosstalk results in the recall of undesired attractor states of the network [26–29].

37th Conference on Neural Information Processing Systems (NeurIPS 2023).

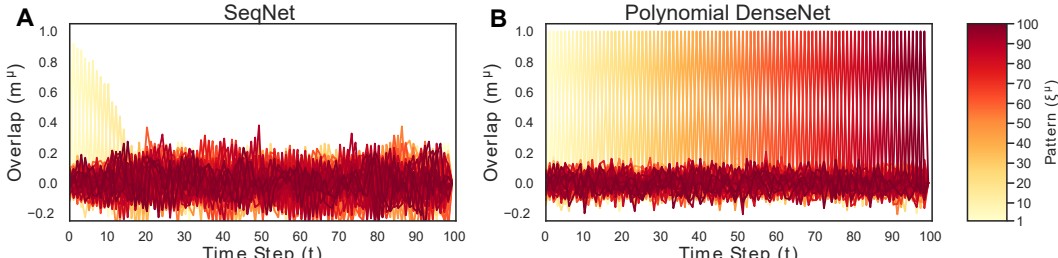

Figure 1: `SeqNet` and Polynomial `DenseNet` ($d = 2$) are simulated with $N = 300$ neurons and $P = 100$ patterns. One hundred curves are plotted as a function of time, each representing the overlap of the network state at time $t$ with one of the patterns, $m^\mu = (1/N) \sum_{i=1}^{N} \xi_i^\mu S_i$. The curves are ordered using the color code described on the right (patterns in the beginning and end of the sequence are shaded in yellow and red respectively). **A**. `SeqNet` quickly loses the correct sequence, indicated by the lack of alignment of the network state with the correct pattern in the sequence ($m^\mu \ll 1$). **B**. The Polynomial `DenseNet` faithfully recalls the entire sequence and maintains alignment with one of the patterns at any moment in time, $m^\mu \approx 1$.

Recent modifications of the Hopfield Network, known as Dense Associative Memories or Modern Hopfield Networks (MHNs), overcome this limitation by introducing a strong nonlinearity when computing the overlap between the state of the network and memory patterns stored in the network [30, 31]. This leads to greater separation between partially overlapping memories, thereby reducing crosstalk, increasing the signal-to-noise ratio, and increasing the probability of successful recall [32].

Most models based on the Hopfield Network are autoassocative, meaning they are designed for the robust storage and recall of individual memories. Thus, they are incapable of storing sequences of memories. In order to adapt these models to store sequences, one must utilize asymmetric weights in order to drive the network from one activity pattern to the next. Many such models use temporally asymmetric Hebbian learning rules to strengthen synaptic connections between neural activity at one time state and the next time state, thereby learning temporal association between patterns in a sequence [1, 3, 10, 11, 16, 17, 22].

In this paper, we extend Dense Associative Memories to the setting of asymmetric weights in order to store and recall long sequences of memories. We work directly with the update rule for the state of the network, allowing us to provide an analytical derivation for the sequence capacity of our proposed network. We find a close match between theoretical calculation and numerical simulation, and further establish the ability of this model to store and recall sequences of correlated patterns. Additionally, we examine the dynamics of a model containing both symmetric and asymmetric terms. Finally, we describe applications of our network as a model of biological motor control.

## 2   `DenseNet`s for Sequence Storage

Traditional Hopfield Networks and MHNs, as described in Appendix B, are capable of storing individual memories. What about storing sequences? Assume that we want to store a sequence of $P$ patterns, $\boldsymbol{\xi}^1 \to \boldsymbol{\xi}^2 \to \cdots \to \boldsymbol{\xi}^P$, where $\xi_j^\mu \in \{\pm 1\}$ is the $j^{th}$ neuron of the $\mu^{th}$ pattern and the network will transition from pattern $\boldsymbol{\xi}^\mu$ to $\boldsymbol{\xi}^{\mu+1}$. Let $N$ be the number of neurons in the network and $\mathbf{S}(t) \in \{-1, +1\}^N$ be the state of the network at time $t$. We want to design a network with dynamics such that when the network is initialized in pattern $\boldsymbol{\xi}^1$, it will traverse the entire sequence.[1] We define a network, `SeqNet`, which follows a discrete-time synchronous update rule[2]:

$$T_{SN}(\mathbf{S})_i := \text{sgn}\left[\sum_{j \neq i} J_{ij} S_j\right] = \text{sgn}\left[\sum_{\mu=1}^{P} \xi_i^{\mu+1} m_i^\mu\right], \quad m_i^\mu := \frac{1}{(N-1)} \sum_{j \neq i} \xi_j^\mu S_j, \quad (1)$$

---

[1] We impose periodic boundary conditions and define $\boldsymbol{\xi}^{P+1} \equiv \boldsymbol{\xi}^1$. Boundary terms have a sub-leading contribution to the crosstalk, so a model with open boundary conditions will have the same scaling of capacity.

[2] One can also consider an asynchronous update rule in which one neuron is updated at a time [23, 26].

where $\mathbf{S}(t+1) = T_{SN}(\mathbf{S})$ and $J_{ij} = \frac{1}{N}\sum_{\mu=1}^{P} \xi_i^{\mu+1}\xi_j^{\mu}$ is an asymmetric matrix connecting pattern $\xi^{\mu}$ to $\xi^{\mu+1}$. Note that we are excluding self-interaction terms $i = j$. We also rewrote the dynamics in terms of $m_i^{\mu}$, the overlap of the network state $\mathbf{S}$ with pattern $\xi^{\mu}$. When the network is aligned most closely with pattern $\xi^{\mu}$, the overlap $m_i^{\mu}$ is the largest contribution in the sum and pushes the network to pattern $\xi^{\mu+1}$. When multiple patterns have similar overlaps, meaning they are correlated, then there will be low signal-to-noise ratio. This correlation between patterns limits the capacity of the network, limiting the `SeqNet`'s capacity to scale linearly relative to network size.

To overcome the capacity limitations of the `SeqNet`, we use inspiration from Dense Associative Memories [30] to define the `DenseNet` update rule:

$$T_{DN}(\mathbf{S})_i := \mathrm{sgn}\left[\sum_{\mu=1}^{P} \xi_i^{\mu+1} f\left(m_i^{\mu}\right)\right] \tag{2}$$

where $f$ is a nonlinear monotonically increasing interaction function. Similar to MHNs, $f$ reduces the crosstalk between patterns and, as we will analyze in detail, leads to improved capacity. Figure 1 demonstrates this improvement for $f(x) = x^2$.

## 2.1 Sequence capacity

To derive analytical results for the capacity, we must choose a distribution to generate the patterns. As is standard in studies of the classic HN and MHNs [26–31, 33–36], we choose this to be the Rademacher distribution, where $\xi_j^{\mu} \in \{-1, +1\}$ with equal probability for all neurons $j$ in all patterns $\mu$, and calculate the capacity for different update rules. If one is allowed to specially engineer the patterns, even the `SeqNet`can store a sequence of length $2^N$ [37], but this construction is not relevant to associative recall of realistic sequences. Rademacher patterns are a more appropriate model for generic patterns while remaining theoretically tractable.

We consider both the robustness of a single transition, and the robustness of propagation through the full sequence. For a fixed network size $N \in \{2, 3, \ldots\}$ and an error tolerance $c \in [0, 1)$, we define the single-transition and sequence capacities by

$$P_T(N, c) = \max\left\{P \in \{2, \ldots, 2^N\} : \mathbb{P}\left[\mathbf{T}_{DN}(\xi^1) = \xi^2\right] \geq 1 - c\right\} \tag{3}$$

and

$$P_S(N, c) = \max\left\{P \in \{2, \ldots, 2^N\} : \mathbb{P}\left[\cap_{\mu=1}^{P}\{\mathbf{T}_{DN}(\xi^{\mu}) = \xi^{\mu+1}\}\right] \geq 1 - c\right\}, \tag{4}$$

respectively, where the probability is taken over the random patterns. Note that for the single-transition capacity we could focus on any pair of subsequent patterns due to translation invariance arising from periodic boundary conditions. Also note that the full sequence capacity is defined by demanding that all transitions are correct. For perfect recall, we want to take the threshold $c \downarrow 0$. In the thermodynamic limit in which $N, P \to \infty$, we expect for there to exist a sharp transition in the recall probabilities as a function of $P$, with almost-surely perfect recall below the threshold value and vanishing probability of recall above [26–29, 31, 33–36]. Thus, we expect the capacity to become insensitive to the value of $c$ in the thermodynamic limit; this is known rigorously for the classic Hopfield network from the work of Bovier [34].

As we detail in Appendix C, all of our theoretical results are obtained under two approximations. We will validate the accuracy of the resulting capacity predictions through comparison with numerical experiments. First, following Petritis [33]'s analysis of the classic Hopfield network, we use union bounds to control the single-transition and full-sequence capacities in terms of the single-bitflip error probability $\mathbb{P}[T_{DN}(\xi^1)_1 \neq \xi_1^2]$. Using the fact that the patterns are i.i.d., this gives $\mathbb{P}[\mathbf{T}_{DN}(\xi^{\mu}) = \xi^{\mu+1}] \geq 1 - N\mathbb{P}[T_{DN}(\xi^1)_1 \neq \xi_2^1]$ and $\mathbb{P}[\cap_{\mu=1}^{P}\{\mathbf{T}_{DN}(\xi^{\mu}) = \xi^{\mu+1}\}] \geq 1 - NP\mathbb{P}[T_{DN}(\xi^1)_1 \neq \xi_2^1]$, respectively, resulting in the lower bounds

$$P_T(N, c) \geq \max\left\{P \in \{2, \ldots, 2^N\} : N\mathbb{P}[T_{DN}(\xi^1)_1 \neq \xi_2^1] \leq c\right\}, \tag{5}$$

$$P_S(N, c) \geq \max\left\{P \in \{2, \ldots, 2^N\} : NP\mathbb{P}[T_{DN}(\xi^1)_1 \neq \xi_2^1] \leq c\right\}. \tag{6}$$

From studies of the classic Hopfield network, we expect for these bounds to be tight in the thermodynamic limit ($N \to \infty$), but we will not attempt to prove that this is so [33, 34]. Second, our theoretical

results are obtained under the approximation of $\mathbb{P}[T_{HN}(\boldsymbol{\xi}^1)_1 \neq \xi_1^2]$ in the regime $N, P \gg 1$ by a Gaussian tail probability. Concretely, we write the single-bitflip probability as

$$\mathbb{P}[T_{DN}(\boldsymbol{\xi}^1)_1 \neq \xi_1^2] = \mathbb{P}[C < -f(1)] \tag{7}$$

in terms of the crosstalk

$$C = \sum_{\mu=2}^{P} \xi_1^2 \xi_1^{\mu+1} f\left(\frac{1}{N-1} \sum_{j=2}^{N} \xi_j^\mu \xi_j^1\right), \tag{8}$$

which represents interference between patterns that can lead to a bitflip. Then, as the crosstalk is the sum of $P - 1$ i.i.d. random variables, we approximate its distribution as Gaussian. We then extract the capacity by determining how $P$ should scale with $N$ such that the error probability tends to zero as $N \to \infty$, corresponding to taking $c \downarrow 0$ with increasing $N$. Within the Gaussian approximation, we can also estimate the capacity at fixed $c$ by using the asymptotics of the inverse Gaussian tail distribution function to determine how $P$ should scale with $N$ such that the error probability is asymptotically bounded by $c$ as $N \to \infty$. This predicts that the effect of non-negligible $c$ should vanish as $N \to \infty$.

For $P$ large but finite, this Gaussian approximation amounts to retaining only the leading term in the Edgeworth expansion of the tail distribution function [38–41]. We will not endeavour to rigorously control the error of this approximation in the regime of interest in which $N$ is also large. To convert our heuristic results into fully rigorous asymptotics, one would want to construct an Edgeworth-type series expansion for the tail probability $\mathbb{P}[C < -f(1)]$ that is valid in the joint limit with rigorously-controlled asymptotic error, accounting for the fact that the crosstalk is a sum of discrete random variables [38–41]. As a simple probe of Gaussianity, we will consider the excess kurtosis of the crosstalk distribution, which determines the leading correction to the Gaussian approximation in the Edgeworth expansion, and describes whether its tails are heavier or narrower than Gaussian [38–41].

## 2.2 Polynomial `DenseNet`

Consider the `DenseNet` with polynomial interaction function, $f(x) = x^d$, which we will call the Polynomial `DenseNet`. In Appendix C.1, we argue that the leading asymptotics of the transition and sequence capacities for perfect recall are given by

$$P_T \sim \frac{N^d}{2(2d-1)!! \log(N)}, \quad P_S \sim \frac{N^d}{2(d+1)(2d-1)!! \log(N)}. \tag{9}$$

Note that this polynomial scaling of the single-transition capacity with network size coincides with the capacity scaling of the symmetric MHN [30]. Indeed, as we have excluded self-interaction terms in the update rule, the single-bitflip probabilities for these two models coincide exactly for unbiased Radamacher patterns (Appendix C.1). This allows us to adapt arguments from Demircigil et al. [31] to show that (9) is in fact a rigorous asymptotic lower bound on the capacity (Appendix D). We compare our results for the single-transition and sequence capacities to numerical simulation in Figure 2. The simulation matches theoretical prediction for large network size $N$. For smaller $N$, there are finite-size effects that result in deviation from theoretical prediction. The crosstalk has non-negligible kurtosis in finite size networks which leads to deviation from the Gaussian approximation.

Furthermore, we point out that for fixed $N$, the network capacity does not monotonically increase in the degree $d$. Since the factorial function grows faster than the exponential function, every finite network of size $N$ has a polynomial degree $d_{max}$ after which the capacity will actually decrease. This is also true for the standard MHN. We demonstrate this numerically in Figure 2B, again noting mild deviations between theory and simulation due to finite-size effects.

## 2.3 Exponential `DenseNet`

We have shown the `DenseNet`'s capacity can scale polynomially with network size. Can it scale exponentially? Consider the `DenseNet` with exponential interaction function, $f(x) = e^{(N-1)(x-1)}$, which we call the Exponential `DenseNet`. This function reduces crosstalk dramatically: $f(m^\mu(\mathbf{S})) = 1$ when $m^\mu(\mathbf{S}) = 1$ and is otherwise sent to zero exponentially fast. In Appendix C.2, we show that under the abovementioned approximations one has the leading asymptotics

$$P_T \sim \frac{\beta^{N-1}}{2 \log N} \quad \text{and} \quad P_S \sim \frac{\beta^{N-1}}{2 \log(\beta) N}, \quad \text{where} \quad \beta = \frac{\exp(2)}{\cosh(2)} \simeq 1.964\ldots \tag{10}$$

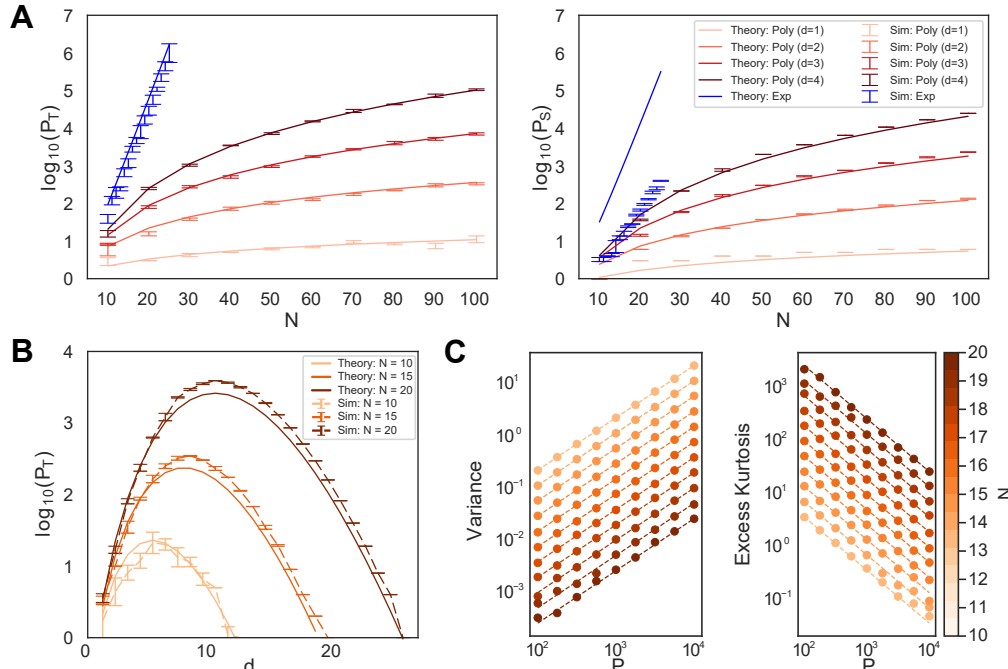

Figure 2: Testing the transition and sequence capacities of `DenseNets` with polynomial and exponential nonlinearities. **A**. Scaling of transition capacity ($\log_{10}(P_T)$, *left*) and sequence capacity ($\log_{10}(P_S)$, *right*) with network size. As network size increases, the variance of the crosstalk decreases and the theoretical approximations become more accurate, resulting in a tight match between theory (solid lines) and simulation (points with error bars). The theory curves are given by Equations 9 and 10. Error bars are computed across realizations of the random patterns (see Appendix G). There is significant deviation between theory and simulation for the sequence capacity of the Exponential `DenseNet`. We show that this is due to finite-size effects in Section 2.3. **B**. Transition capacity of Polynomial `DenseNets` as a function of degree. For any finite network size $N$, there is a degree $d$ that maximizes the transition capacity. The same would be true for the sequence capacity. **C**. Crosstalk variance (*left*) and excess kurtosis (*right*) for the Exponential `DenseNet` as a function of $P$ and $N$. Variance is proportional to $P$ and inversely proportional to $N$, while the opposite is true for excess kurtosis. See Appendix G for details of our numerical methods.

In Figure 2, numerical simulations confirm this model scales significantly better than the Polynomial `DenseNet` and enables one to store exponentially long sequences relative to network size. While the ratio between transition and sequence capacities remains bounded for the Polynomial `DenseNet`, where $P_T/P_S \sim d + 1$, the gap for the Exponential `DenseNet` diverges with network size.

However, we can see in Figure 2A that the empirically measured capacity—particularly the sequence capacity—of the Exponential `DenseNet` deviates substantially from the predictions of our approximate Gaussian theory. Due to computational constraints, our numerical simulations are limited to small network sizes (Appendix G). Computing the excess kurtosis of the crosstalk distribution with a number of patterns comparable to the capacity predicted by the Gaussian theory reveals that, for the range of system sizes we can simulate, the distribution should deviate strongly from a Gaussian. In particular, if take $P \sim \beta^{N-1}/(\alpha N)$ for some constant factor $\alpha$, then the excess kurtosis increases with network size up to around $N \approx 56$ (Appendix C.2). Increasing the size of an Exponential `DenseNet` therefore has competing effects: for a fixed sequence length $P$, increasing network size $N$ decreases the crosstalk variance, which should reduce the bitflip probability, but also increases the excess kurtosis, which reflects a fattening of the crosstalk distribution tails that should increase the bitflip probability. This is illustrated in Figure 2C.

The competition between increasing $P$ and $N$ for the Exponential `DenseNet` is easy to understand intuitively. For a fixed $N$, increasing $P$ means that the crosstalk is equal in distribution to the sum of an increasingly large number of i.i.d. random variables, and thus by the central limit theorem should become increasingly Gaussian. Conversely, for a fixed $P$, increasing $N$ means that each of the

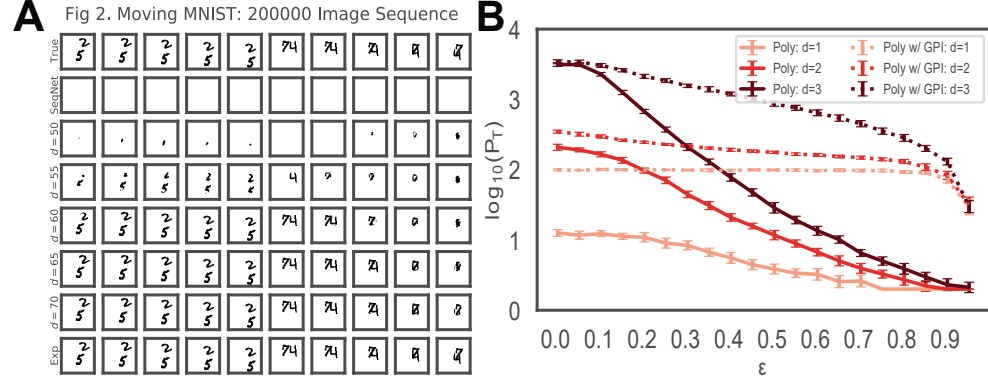

Figure 3: **A**. Recall of a sequence of 200000 correlated images from the MovingMNIST dataset using `DenseNets` of size $N = 784$. We showcase a 10 image subsequence. The top row depicts the true sequence, the second row depicts `SeqNet`'s performance, the next rows depict the Polynomial `DenseNets`' performance which increases with degree $d$, and the final row depicts the Exponential `DenseNet`'s performance which yields perfect recall. **B**. Transition capacity of Polynomial `DenseNets` of size $N = 100$ relative to pattern bias $\epsilon$. Increasing $\epsilon$ monotonically decreases capacity. Networks with stronger nonlinearities maintain high capacity for large correlation strength. Implementing the generalized pseudoinverse rule decorrelates these patterns and maintains high sequence capacity for much larger correlation. See Appendix G for details of numerical methods.

$P - 1$ contributions to the crosstalk is equal in distribution to the *product* of an increasing number of i.i.d. random variables—as $f\left(\frac{1}{N-1} \sum_{j=2}^{N} \xi_j^\mu \xi_j^1\right) = \prod_{j=2}^{N} \exp(\xi_j^\mu \xi_j^1)$—and thus by the multiplicative central limit theorem each term should tend to a lognormal distribution. In this regime, then, the crosstalk is roughly a mixture of lognormals, which is decidedly non-Gaussian. In contrast, for a Polynomial `DenseNet`, memorization is easy in the limit where $N$ tends to infinity for fixed $P$, as the crosstalk should tend almost surely to zero as each term $f\left(\frac{1}{N-1} \sum_{j=2}^{N} \xi_j^\mu \xi_j^1\right) \to 0$ almost surely.

## 2.4 Recalling Sequences of Correlated Patterns

The full-sequence capacity scaling laws for these models were derived under the assumption of i.i.d Rademacher random patterns. While theoretically convenient, this is unrealistic for real-world data. We therefore test these networks in more realistic settings by storing correlated sequences of patterns, which will lead to greater crosstalk in each transition and thus smaller single-transition and full-sequence capacities relative to network size [26, 36]. However, the nonlinear interaction functions should still assist in separating correlated patterns to enable successful sequence recall.

For demonstration, we store a sequence of 200000 highly-correlated images from the MovingMNIST dataset and attempt to recall this sequence using `DenseNets` with different nonlinearities [42]. The entire sequence is composed of 10000 unique subsequences concatenated together, where each subsequence is composed of 20 images of two hand-written digits slowly moving through one another. This means there is significant correlation between patterns which will result in large amounts of crosstalk. The results of the `DenseNets` are shown in Figure 3A, where increasing the nonlinearity of the Polynomial `DenseNets` slowly improves recall but not entirely, while the exponential network achieves perfect recall. The `SeqNet` and `DenseNets`, up until approximately $d = 50$, are entirely unable to recall any part of any image, despite the `DenseNets` being well within the capacity limits predicted by theoretical calculations on uncorrelated patterns.

## 2.5 Generalized pseudoinverse rule

Can we overcome the `DenseNet`'s limited ability to store correlated patterns? Drawing inspiration from the pseudoinverse learning rule introduced by Kanter and Sompolinsky [43] for the classic Hopfield network, we propose a generalized pseudoinverse (GPI) transition rule

$$T_{GPI}(\mathbf{S})_i := \text{sgn}\left[\sum_{\mu=1}^{P} \xi_i^{\mu+1} f\left(\sum_{\nu=1}^{P} (O^+)^{\mu\nu} m^\nu(\mathbf{S})\right)\right], \quad O^{\mu\nu} = \frac{1}{N} \sum_{j=1}^{N} \xi_j^\mu \xi_j^\nu, \qquad (11)$$

where the overlap matrix $O^{\mu\nu}$ is positive-semidefinite, so we can define its pseudoinverse $\mathbf{O}^+$ by inverting the non-zero eigenvalues. With $f(x) = x$, this reduces to the pseudoinverse rule of [43].

If the patterns are linearly independent, such that $\mathbf{O}$ is full-rank, we can see that this rule can perfectly recall the full sequence (Appendix E). This matches the classic pseudoinverse rule's ability to perfectly store any set of linearly independent patterns; this is why we choose to sum over $\nu$ inside the separation function in (11). For i.i.d. Rademacher patterns, linear independence holds almost surely in the thermodynamic limit provided that $P < N$.

In Figure 3B, we demonstrate the effect of correlation on the Polynomial `DenseNet` through studying the recall of biased patterns $\xi_i^\mu$ with $\mathbb{P}(\xi_i^\mu = \pm 1) = \frac{1}{2}(1 \pm \epsilon)$ for $\epsilon \in [0, 1)$.[3] We see that the Polynomial `DenseNet` has better recall at all levels of bias $\epsilon$ as degree $d$ increases, although we still expect there to be a maximum degree as described before. However, at large correlation values, they all have low recall, suggesting the need for alternative methods to decorrelate these patterns. This failure is easy to understand theoretically, following van Hemmen and Kühn [44]'s analysis of the classic Hopfield model: for patterns with bias $\epsilon$, the Polynomial `DenseNet` update rule expands as

$$T_{DN}(\boldsymbol{\xi}^\mu)_i = \text{sgn}[\xi_i^{\mu+1} + (P-1)\epsilon^{2d+1} + \mathcal{O}(\sqrt{P/N})]. \tag{12}$$

Therefore, even if $N$ is large, for $\epsilon \neq 0$ there must be some value of $P$ for which the constant bias overwhelms the signal. If $N \to \infty$ for any fixed $P$, then we must have $P < \epsilon^{-(2d+1)} + 1$ for the signal to dominate. In Figure 3B, we show the generalized pseudoinverse update rule is more robust to large correlations than the Polynomial `DenseNet`. While this rule can also be applied to the Exponential `DenseNet`, simulations fail due to numerical instability coming from small values in the pseudoinverse.

## 3 `MixedNet`s for variable timing

Thus far, we have considered sequence recall in purely asymmetric networks. These networks transition to the next pattern in the sequence at every timestep, preventing the network from storing sequences with longer timing between elements. In this section, we aim to construct a model where the network stays in a pattern for $\tau$ steps. Our starting model will be an associative memory model for storing sequences known as the Temporal Association Network (TAN) [1, 10], defined as:

$$T_{TAN}(\mathbf{S})_i := \text{sgn}\left[\sum_{\mu=1}^{P}\left[\xi_i^\mu m_i^\mu + \lambda\xi_i^{\mu+1}\bar{m}_i^\mu\right]\right], \quad \bar{m}_i^\mu := \frac{1}{N-1}\sum_{j\neq i}\xi_j^\mu\bar{S}_j \tag{13}$$

where $\bar{m}_i^\mu$ represents the normalized overlap of each pattern $\boldsymbol{\xi}^\mu$ with a weighted time-average of the network over the past $\tau$ timesteps, $\bar{S}_i(t) = \sum_{\rho=0}^{\tau} w(\rho)S_i(t-\rho)$. The weight function, $w(t)$, is generally taken to be a low-pass convolutional filter (e.g. Heaviside step function, exponential decay).

This network combines a symmetric and asymmetric term for robust recall of multiple sequences. The symmetric term containing $m_i^\mu(t)$, also referred to as a "fast" synapse, stabilizes the network in pattern $\boldsymbol{\xi}^\mu$ for a desired amount of time. The asymmetric term containing $\bar{m}_i^\mu(t)$, also referred to as a "slow" synapse, drives the network transition to pattern $\boldsymbol{\xi}^{\mu+1}$. The $\lambda$ parameter controls the strength of the transition signal. If $\lambda$ is too small, no transitions will occur since the symmetric term will overpower it. If $\lambda$ is too large, transitions will occur too quickly for the network to stabilize in a desired pattern and the sequence will quickly destabilize.

For TAN, Sompolinsky and Kanter [10] used numerical simulations to estimate the capacity as approximately $P_{TAN} \sim 0.1N$, defining capacity as the ability to recall the sequence in correct order with high overlap (meaning that a small propotion of incorrect bits are allowed in each transition). Note that this model can fail in two ways: (i) it can fail to recall the correct sequence of patterns, or (ii) it can fail to stay in each state for the desired amount of time.

To address these issues, we consider the following dynamics:

$$T_{MN}(\mathbf{S})_i := \text{sgn}\left[\sum_{\mu=1}^{P}\left[\xi_i^\mu f_S\left(m_i^\mu\right) + \lambda\xi_i^{\mu+1}f_A\left(\bar{m}_i^\mu\right)\right]\right] \tag{14}$$

---

[3]At $\epsilon = 1$, the patterns will be deterministic with $\xi_i^\mu = +1$.

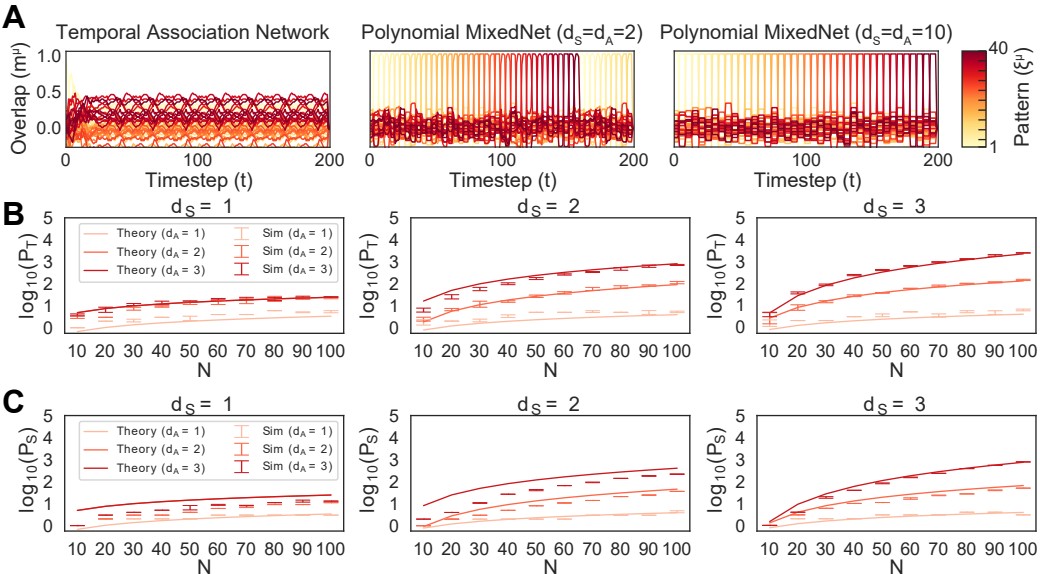

Figure 4: Capacity of the Polynomial `MixedNet`. **A.** We simulate `MixedNets` with $N = 100$, $\tau = 5$, and attempt to store $P = 40$ patterns. The Temporal Association Network (*left*), corresponding to a linear `MixedNet` with $d_S = 1 = d_A$, fails to recover the sequence. Increasing the nonlinearities to $d_S = 2 = d_A$ (*center*) recovers the correct sequence order, but not the timing. Increasing the nonlinearities to $d_S = 10 = d_A$ (*right*) recovers the correct sequence order and timing. **B.** Transition capacity $\log_{10}(P_T)$ of the Polynomial `MixedNet` as a function of network size. Each panel has a fixed symmetric nonlinearity $f_S(x) = x^{d_S}$ indicated by the panel's title. As network size increases, crosstalk variance decreases and theoretical approximations in Equation 3 become more accurate to tightly match the simulations. Note that as expected, the capacity scales according to the minimum of $d_S$ and $d_A$. **C.** As in **B**, but for the sequence capacity $\log_{10}(P_S)$.

We call this model the `MixedNet`, and seek to analyze the relationship between the symmetric and asymmetric terms in driving network dynamics and their impact on sequence capacity. As before, the asymmetric term will try to push the network to the next state at every timestep, while the symmetric term tries to maintain it in its current state for $\tau$ timesteps. We will allow different nonlinearities for $f_S$ and $f_A$, and analyze their effect on transition and sequence capacity.

We demonstrate the effectiveness of the Polynomial `MixedNet`, where for simplicity we set $f_S(x) = f_A(x) = x^d$, in Figure 4A. While TAN fails completely, a polynomial nonlinearity of $d = 2$ enables recall of pattern order but the network does not stay in each pattern for $\tau = 5$ timesteps. Further increasing the nonlinearity to $d = 10$ recovers the desired sequence with correct order and timing.

Theoretical analysis of the capacity of the `MixedNet` (14) for general memory length $\tau$ is challenging due to the extended temporal interactions. We therefore consider single-step memory ($\tau = 1$), and show that even in this relatively tractable special case new complications arise relative to our analysis of the `DenseNet`. Alternatively, we can interpret the `MixedNet` with $\tau = 1$ as an imperfectly-learned `DenseNet`. If one imagines the network learns its weights through a temporally asymmetric Hebbian rule with an extended plasticity kernel, and its state is not perfectly clamped to the desired transition, the coupling from $\boldsymbol{\xi}^\mu$ to $\boldsymbol{\xi}^{\mu+1}$ could be corrupted by coupling $\boldsymbol{\xi}^\mu$ to itself [22].

We first consider the setting where both interaction functions are polynomial, $f_S(x) = x^{d_S}$ and $f_A(x) = x^{d_A}$, and refer to this network as the Polynomial `MixedNet`. This model is analyzed in detail in Appendix F.1. Interestingly, this model's crosstalk variance forms a bimodal distribution, as shown in Figure F.1. This complicates the analysis, but once bimodality is accounted for one can approximate the capacity using a similar argument to that of the `DenseNet`. We find that

$$P_T \sim \frac{(\lambda - 1)^2}{2\gamma_{d_S, d_A}} \frac{N^{\min\{d_S, d_A\}}}{\log N}, \quad P_S \sim \frac{(\lambda - 1)^2}{2(\min\{d_S, d_A\} + 1)\gamma_{d_S, d_A}} \frac{N^{\min\{d_S, d_A\}}}{\log N}, \quad (15)$$

where $\gamma_{d_S,d_A}$ is a multiplicative factor defined as

$$\gamma_{d_S,d_A} = \begin{cases} (2d_S - 1)!! & \text{, if } d_S < d_A \\ (\lambda^2 + 1)(2d_S - 1)!! + 2\lambda[(d_S - 1)!!]^2 \mathbf{1}\{d_S \text{ even}\} & \text{, if } d_S = d_A \\ \lambda^2(2d_A - 1)!! & \text{, if } d_S > d_A. \end{cases} \quad (16)$$

In Figure 4B-C, we show that simulations match the theory curves well as $N$ increases. We demonstrate theoretical and simulations results for the Exponential `MixedNet` in Appendix F.2.

## 4    Biologically-Plausible Implementation

Since biological neural networks must store sequence memories [2, 5–8], one naturally asks if these results can be generalized to biologically-plausible neural networks. A straightforward biological interpretation of the `DenseNet` is problematic, as a network with polynomial interaction function of degree $d$ is equivalent to having a neural network with many-body synapses between $d + 1$ neurons. This can be seen by expanding the Polynomial `DenseNet` in terms of a weight tensor of $d+1$ neurons:

$$S_i(t + 1) = \text{sgn}\left[\sum_{j_1,\dots,j_d} J_{ij_1\dots j_d} S_{j_1}(t)\dots S_{j_d}(t)\right], \quad J_{i,j_1,\dots,j_d} = \frac{1}{N^d}\sum_{\mu=1}^P \xi_i^{\mu+1}\xi_{j_1}^\mu\cdots\xi_{j_d}^\mu \quad (17)$$

This is biologically unrealistic as synaptic connections usually occur between two neurons [45]. In the case of the Exponential `DenseNet`, one can interpret its interaction function via a Taylor series expansion, implying synaptic connections between infinitely many neurons which is even more problematic. Similar difficulties arise in models with sum of terms with different powers [46].

To address this issue, we again take inspiration from earlier work in MHNs. Krotov and Hopfield [47] addressed this concern for symmetric MHNs by reformulating the network using two-body synapses, where the network was partitioned into a bipartite graph with visible and hidden neurons (see [48] for an extension of this idea to deeper networks). The visible neurons correspond to the neurons in our network dynamics, $\mathbf{S}_j$, while the hidden neurons correspond to the individual memories stored within the network. They are connected through a weight matrix. Since we are working with an asymmetric network, we modify their approach and define two sets of synaptic weights: $W_{j\mu}$ connects visible neuron $v_j$ to hidden neuron $h_\mu$, $M_{\mu j}$ connects hidden neuron $h_\mu$ to visible neuron $v_j$. This yields the same dynamics exhibited in Equation (2), absorbing the nonlinearity into the hidden neurons' dynamics.

$$h_\mu(t) = f\left(\sum_j W_{j\mu}v_j(t)\right)$$

$$W_{j\mu} = \frac{1}{N}\xi_j^\mu \qquad M_{\mu j} = \xi_j^{\mu+1}$$

$$v_j(t+1) = \text{sgn}\left[\sum_\mu M_{\mu j}h_\mu(t)\right]$$

Figure 5: Biologically-plausible implementation of `DenseNet` with two-body synapses.

For the `DenseNet`, we define the weights as $W_{j\mu} := \frac{1}{N}\xi_j^\mu$ and $M_{\mu j} := \xi_j^{\mu+1}$. For the `MixedNet`, we redefine the weight matrix $M_{\mu j} = \xi_j^\mu + \lambda\xi_j^{\mu+1}$. The update rules for the neurons are as follows:

$$h_\mu(t) := f\left[\sum_j W_{j\mu}v_j(t)\right], \qquad v_j(t+1) := \text{sgn}\left[\sum_\mu M_{\mu j}h_\mu(t)\right] \quad (18)$$

Note that these networks' transition and sequence capacities, $P_T$ and $P_S$, now scale linearly with respect to the total number of neurons in this model, $N$ visible neurons and $P$ hidden neurons. However, the network capacity still scales nonlinearly with respect to the number of visible neurons.

Finally, we remark that this network is reminiscent of recent computational models for motor action selection and control via the cortico-basal ganglia-thalamo-cortical loop, in which the basal ganglia inhibits thalamic neurons that are bidirectionally connected to a recurrent cortical network [5, 49, 50]. This relates to our model as follows: the motor cortex (visible neurons) executes an action, each

thalamic unit (hidden neurons) encodes a motor motif, and the basal ganglia silences thalamic neurons (external network modulating context). In particular, the role of the basal ganglia in this network suggests a novel mechanism of context-dependent gating within Hopfield Networks [51]. Rather than modulating synapses or feature neurons in a network, one can directly inhibit (activate) memory neurons in order to decrease (increase) the likelihood of transitioning to the associated state. Similarly, thalamocortical loops have been found to be important to song generation in zebra finches [52]. Thus, the biological implementation of the `DenseNet` can provide insight into how biological agents reliably store and generate complex sequences.

## 5  Discussion and Future Directions

We introduced the `DenseNet` for the reliable storage and recall of long sequences of patterns, derived the scaling of its single-transition and full-sequence capacity, and verified these results in numerical simulation. We found that depending on the choice of nonlinear interaction function, the `DenseNet` could scale polynomially or exponentially. We tested the ability of these models to recall sequences of correlated patterns, by comparing the recall of a sequence of MovingMNIST images with different nonlinearities. As expected, the network's reconstruction capabilities increased with the nonlinearity power $d$, with perfect recall achieved by the exponential nonlinearity. To further increase the capacity, we introduced the generalized pseudoinverse rule and demonstrated in simulation its ability to maintain high capacity for highly correlated patterns. We also introduced and analyzed the `MixedNet` to maintain patterns within sequences for longer periods of time. Finally, we described a biologically plausible implementation of the models with connections to motor control.

There has recently been a renewed interest in storing sequences of memories. Steinberg and Sompolinsky [53] store sequences in Hopfield networks by using a vector-symbolic architecture to bind each pattern to its temporal order in the sequence, thus storing the entire sequence as a single attractor. However, this model suffers from the same capacity limitations as the Hopfield Network. Whittington et al. [54] suggest a mechanism to control sequence retrieval via an external controller, analogous to the role we ascribe to the basal ganglia for context-dependent gating. Herron et al. [55] investigate a mechanism for robust sequence recall within complex systems more broadly, reducing crosstalk by directly modulating interactions between neurons rather than the inputs into neurons. Tang et al. [56] propose a model for sequential recall akin to `SeqNet` with an implicit statistical whitening process. Karuvally et al. [57] introduce a model closely related to the biologically-plausible implementation of our `MixedNet` and analyze it in the setting of continuous-time dynamics, allowing for intralayer synapses within the hidden layer and different timescales between the hidden and feature layers.

While we have focused on a generalization of the fixed-point capacity for sequence memory, this is not the only notion of capacity one could consider. In other studies of MHNs, instead of considering stability as the probability of staying at a fixed point, researchers quantify the probability that the network will reach a fixed point within a single transition [31, 58, 59]. This approach allows one to quantify noise-robustness and the size of each memory's basin of attraction [35]. More broadly, one could consider other definitions of associative memory capacity not addressed here, including those that depend only on network architecture and not on the assumption of a particular learning rule [60, 61]. However, as compared to the relatively simple analysis that is possible for the fixed-point capacity of a Hopfield network using a Hebbian learning rule, analyzing these alternative notions of capacity in nonlinear networks can pose significant technical challenges [61–63].

In this work, we limited ourselves to theoretical analysis of discrete-time networks storing binary patterns. An important direction for future research would be to go beyond the Gaussian theory in order to develop accurate predictions of the Exponential `DenseNet` capacity. There are also many potential avenues for extending these models and methods to continuous-time networks, continuous-valued patterns, computing capacity for correlated patterns, testing different weight functions, and examining different network topologies. Finally, we hope to take inspiration from the recent resurgence of RNNs in long sequence modeling to use this model for real-world tasks [64, 65].

## Acknowledgments and Disclosure of Funding

We thank Matthew Farrell, Shanshan Qin, and Sabarish Sainathan for useful discussions and comments on earlier versions of our manuscript. HC was supported by the GFSD Fellowship, Harvard GSAS Prize Fellowship, and Harvard James Mills Peirce Fellowship. JAZ-V and CP were supported by NSF Award DMS-2134157 and NSF CAREER Award IIS-2239780. CP received additional support from a Sloan Research Fellowship. This work has been made possible in part by a gift from the Chan Zuckerberg Initiative Foundation to establish the Kempner Institute for the Study of Natural and Artificial Intelligence. The computations in this paper were run on the FASRC Cannon cluster supported by the FAS Division of Science Research Computing Group at Harvard University.

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

# A    Review of Modern Hopfield Networks

Here we review the Hopfield network and its modern generalization as an auto-associative memory model. These ideas will be helpful for storing sequences in network dynamics.

## A.1    The Hopfield Network

We first introduce the classic Hopfield Network [23]. Let $N$ be the number of neurons in the network and $\mathbf{S}(t) \in \{-1, +1\}^N$ be the state of the network at time $t$. The task is to store $P$ patterns, $\{\boldsymbol{\xi}^1, \dots, \boldsymbol{\xi}^\mu\}$, where $\xi_j^\mu \in \{\pm 1\}$ is the $j^{th}$ neuron of the $\mu^{th}$ pattern. The goal is to design a network with dynamics such that when the network is initialized with a pattern, it will converge to one of the stored memories.

The Hopfield Network [23] attempts this by following the discrete-time synchronous update rule[4]:

$$\mathbf{S}(t + 1) = \mathbf{T}_{HN}(\mathbf{S}(t)), \tag{A.1}$$

where the transition operator $T_{HN}(\cdot)_i$ for neuron $i$ is defined in terms of symmetric Hebbian weights:

$$T_{HN}(\mathbf{S})_i = \mathrm{sgn}\left[\sum_{j \neq i} J_{ij} S_j\right], \quad J_{ij} = \frac{1}{N} \sum_{\mu=1}^P \xi_i^\mu \xi_j^\mu. \tag{A.2}$$

Note that we are excluding self-interaction terms ($J_{ii}$) in Equation A.2. To interpret this dynamics from another useful point of view, we define the overlap, or Mattis magnetization, $m_i^\mu$ of the network state $\mathbf{S}$ with pattern $\boldsymbol{\xi}^\mu$. We can then rewrite the update rule for the Hopfield Network as

$$T_{HN}(\mathbf{S})_i := \mathrm{sgn}\left[\sum_{\mu=1}^P \xi_i^\mu m_i^\mu\right], \quad m_i^\mu := \frac{1}{(N-1)} \sum_{j \neq i} \xi_j^\mu S_j \tag{A.3}$$

We interpret this as at every time $t$, the network tries to identify the pattern $\boldsymbol{\xi}^\mu$ it is closest to and updates neuron $i$ to the value for that pattern. A natural question to ask about the associative memory networks is their capacity: how many patterns can be stored and recalled with minimal error? This question has been the subject of many studies [23, 27–29, 33–36]. Intuitively, in recalling a pattern $\boldsymbol{\xi}^\nu$, what limits the network's capacity is the overlap between the pattern $\boldsymbol{\xi}^\nu$ and other patterns, referred to as the crosstalk [26, 36].

A precise answer to the storage capacity question can be given under the assumption that the patterns $\{\boldsymbol{\xi}^\mu\}$ are sampled from some probability distribution. While different notions of capacity have been considered in the literature [23, 27–29, 33–36], we focus on the *fixed-point capacity*, which characterizes the probability that, when initialized at a given pattern, the network dynamics do not move the state away from that point. To render the problem analytically tractable, it is usually assumed that the pattern components are i.i.d. Rademacher random variables, i.e., $\mathbb{P}(\xi_j^\mu = \pm 1) = 1/2$ for all $j$ and $\mu$. Then, at finite network size one can define the capacity as

$$P_{HN}(N, c) = \max\left\{P \in \{2, \dots, 2^N\} : \mathbb{P}\left[\cap_{\mu=1}^P \{\mathbf{T}_{HN}(\boldsymbol{\xi}^\mu) = \boldsymbol{\xi}^\mu\}\right] \geq 1 - c\right\}, \tag{A.4}$$

where $c \in [0, 1)$ is a fixed error tolerance. As we review in detail in Appendix B, one finds an asymptotic capacity estimate $P_{HN} \sim \frac{N}{4\log(N)}$ for $c = 0$, which can be shown to be a sharp threshold [33–35].

## A.2    Modern Hopfield Networks

Recent work from Krotov and Hopfield [30, 66] reinvigorated a line of research into generalized Hopfield Networks with larger capacity [67–72], resulting in what are now called Dense Associative Memories or Modern Hopfield Networks:

$$T_{MHN}(\mathbf{S})_i := \mathrm{sgn}\left[\sum_{\mu=1}^P \xi_i^\mu f\left(m_i^\mu\right)\right] \tag{A.5}$$

---

[4]For the Hopfield network, one can also consider an asynchronous update rule in which only one neuron is updated at each timestep [23, 26].

where $f$, referred to as the interaction function, is a nonlinear monotonically increasing function whose purpose is to separate the pattern overlaps for better signal to noise ratio. Since $m_i^\mu(t)$ has a maximum value of 1, this means contributions from patterns with partial overlaps will be reduced by the interaction function. This diminishes the crosstalk and thereby increases the probability of transitioning to the correct pattern. If the interaction function is chosen to be $f(x) = x^d$, then the MHN's capacity has been shown to scale polynomially with network size as $P \sim \beta_d \frac{N^d}{\log(N)}$, where $\beta_d$ is a numerical constant depending on the degree $d$ [30, 73–75]. Using a different definition of capacity, Demircigil et al. [31] have also shown that an exponential nonlinearity can lead to exponential scaling of the capacity. See [32] for a recent review of these results.

## B  Review of Hopfield network fixed-point capacity

In this Appendix, we review the computation of the classical Hopfield network fixed-point capacity. Our approach will follow—but not exactly match—that of Petritis [33]. Though these results are standard, we review them in detail both because this approach will inspire in part our approach to the `DenseNet`, and because several important steps of the analysis are significantly simpler than the corresponding steps for the `DenseNet`.

We begin by recalling that the Hopfield network update can be written as

$$T_{HN}(\mathbf{S})_i := \operatorname{sgn}\left[\sum_{\mu=1}^{P} \xi_i^\mu \left(\frac{1}{N-1}\sum_{j\neq i}\xi_j^\mu S_j\right)\right], \tag{B.1}$$

and that our goal is to determine

$$P_{HN}(N, c) = \max\left\{P \in \{2, \ldots, 2^N\} : \mathbb{P}\left[\cap_{\mu=1}^{P}\{\mathbf{T}_{HN}(\boldsymbol{\xi}^\mu) = \boldsymbol{\xi}^\mu\}\right] \geq 1 - c\right\} \tag{B.2}$$

for some absolute constant $0 \leq c < 1$, at least in the regime where $N, P \gg 1$ [33–36]. As is standard in theoretical studies of Hopfield model capacity [26–29, 33–36], we take in these probabilities the pattern components $\xi_k^\mu$ to be independent and identically distributed Rademacher random variables. We can expand the memorization probability as a union of single-bitflip events:

$$\mathbb{P}\left[\bigcap_{\mu=1}^{P}\{\mathbf{T}_{HN}(\boldsymbol{\xi}^\mu) = \boldsymbol{\xi}^\mu\}\right] = 1 - \mathbb{P}\left[\bigcup_{\mu=1}^{P}\bigcup_{i=1}^{N}\{T_{HN}(\boldsymbol{\xi}^\mu)_i \neq \xi_i^\mu\}\right]. \tag{B.3}$$

This illustrates why analyzing the memorization probability is complicated: the single-pattern events $\mathbf{T}_{HN}(\boldsymbol{\xi}^\mu) = \boldsymbol{\xi}^\mu$ are not independent across patterns $\mu$, and each single-pattern event is itself the intersection of non-independent single-neuron events $T_{HN}(\boldsymbol{\xi}^\mu)_i = \xi_i^\mu$. However, as the single-bitflip probabilities $\mathbb{P}[T_{HN}(\boldsymbol{\xi}^\mu)_j \neq \xi_j^\mu]$ are identical for all $\mu$ and $j$, we can obtain a straightforward union bound

$$\mathbb{P}\left[\bigcap_{\mu=1}^{P}\{\mathbf{T}_{HN}(\boldsymbol{\xi}^\mu) = \boldsymbol{\xi}^\mu\}\right] = 1 - \mathbb{P}\left[\bigcup_{\mu=1}^{P}\bigcup_{i=1}^{N}\{T_{HN}(\boldsymbol{\xi}^\mu)_i \neq \xi_i^\mu\}\right] \tag{B.4}$$

$$\geq 1 - \sum_{\mu=1}^{P}\sum_{i=1}^{N}\mathbb{P}\left[T_{HN}(\boldsymbol{\xi}^\mu)_i \neq \xi_i^\mu\right] \tag{B.5}$$

$$= 1 - NP\mathbb{P}[T_{HN}(\boldsymbol{\xi}^1)_1 \neq \xi_1^1], \tag{B.6}$$

where we focus without loss of generality on the first element of the first pattern. Therefore, if we can control the single-bitflip probability $\mathbb{P}[T_{HN}(\boldsymbol{\xi}^1)_1 \neq \xi_1^1]$, we can obtain a lower bound on the true capacity. In particular,

$$P_{HN}(N, c) \geq \max\left\{P \in \{2, \ldots, 2^N\} : NP\mathbb{P}[T_{HN}(\boldsymbol{\xi}^1)_1 \neq \xi_1^1] \leq c\right\} \tag{B.7}$$

From the definition of the Hopfield network update rule, we have

$$\mathbb{P}[T_{HN}(\boldsymbol{\xi}^1)_1 \neq \xi_1^1] = \mathbb{P}\left\{ \mathrm{sgn}\left[ \frac{1}{N-1} \sum_{\mu=1}^{P} \sum_{j \neq i} \xi_1^\mu \xi_j^\mu \xi_j^1 \right] \neq \xi_1^1 \right\} \tag{B.8}$$

$$= \mathbb{P}\left[ \frac{1}{N-1} \sum_{\mu=1}^{P} \sum_{j \neq i} \xi_1^1 \xi_1^\mu \xi_j^1 \xi_j^\mu < 0 \right] \tag{B.9}$$

$$= \mathbb{P}\left[ C > 1 \right], \tag{B.10}$$

where we have defined

$$C = \frac{1}{N-1} \sum_{\mu=2}^{P} \sum_{j \neq i} \xi_1^1 \xi_1^\mu \xi_j^1 \xi_j^\mu \tag{B.11}$$

and used the fact that the distribution of $C$ is symmetric. $C$ is referred to as the *crosstalk*, because it represents the effect of interference between the first pattern and the other $P-1$ patterns on recall of the first pattern. We can simplify the crosstalk using the fact that, since we have assumed i.i.d. Rademacher patterns, we have the equality in distribution

$$\xi_j^1 \xi_j^\mu \overset{d}{=} \xi_j^\mu \tag{B.12}$$

for all $\mu = 2, \ldots, P$ and $j = 1, \ldots, N$, yielding

$$C \overset{d}{=} \frac{1}{N-1} \sum_{\mu=2}^{P} \sum_{j \neq i} \xi_1^\mu \xi_j^\mu. \tag{B.13}$$

Similarly, we have

$$\xi_1^\mu \xi_j^\mu \overset{d}{=} \xi_j^\mu \tag{B.14}$$

for all $\mu = 2, \ldots, P$ and $j = 2, \ldots, N$, which finally yields

$$C \overset{d}{=} \frac{1}{N-1} \sum_{\mu=2}^{P} \sum_{j \neq i} \xi_j^\mu. \tag{B.15}$$

Therefore, for the classic Hopfield network the crosstalk is equal in distribution to the sum of $(P-1)(N-1)$ i.i.d. Rademacher random variables.

### B.1 Approach 1: Hoeffding's inequality

Now, we can immediately apply Hoeffding's inequality [76], which implies that for any $t > 0$

$$\mathbb{P}\left[ \sum_{\mu=2}^{P} \sum_{k=2}^{N} \xi_k^\mu > t \right] \leq \exp\left( -\frac{1}{2} \frac{t^2}{(P-1)(N-1)} \right). \tag{B.16}$$

We then have that

$$\mathbb{P}\left[ \sum_{\mu=2}^{P} \sum_{k=2}^{N} \xi_k^\mu > N-1 \right] \leq \exp\left( -\frac{1}{2} \frac{N-1}{P-1} \right). \tag{B.17}$$

We then have the bound

$$P_{HN}(N,c) \geq \max\left\{ P \in \{2, \ldots, 2^N\} : NP \exp\left( -\frac{1}{2} \frac{N-1}{P-1} \right) \leq c \right\}. \tag{B.18}$$

We now want to consider the regime $N \gg 1$, and demand that the error probability should tend to zero as we increase $N$. If we substitute in the *Ansatz*

$$P \sim \frac{N}{\alpha \log N}, \tag{B.19}$$

the bound is easily seen to tend to zero for all $\alpha \geq 4$, yielding an estimated capacity of

$$P_{HN} \sim \frac{N}{4 \log N}. \tag{B.20}$$

As this estimates follows from a sequence of lower bounds on the memorization probability, it is a lower bound on the true capacity of the model [33]. However, via a more involved argument that accounts for the associations between the events $\mathbf{T}_{HN}(\boldsymbol{\xi}^{\mu}) = \boldsymbol{\xi}^{\mu}$, it was shown by Bovier [34] to be tight.

For the classical Hopfield network, the single bitflip probability $\mathbb{P}[C > 1]$ is easy to control using elementary concentration inequalities because the crosstalk can be expressed as a sum of $(P - 1)(N - 1)$ i.i.d. random variables. Therefore, we expect the crosstalk to concentrate whenever $N$ or $P$ or both together are large. However, for the DenseNet, we will find in Appendix C that the crosstalk is given as the sum of $P - 1$ i.i.d. random variables, each of which is a nonlinear function applied to the sum of $N - 1$ i.i.d. Rademacher random variables. Naïve application of Hoeffding's inequality is then not particularly useful. We will therefore take a simpler, though less rigorously controlled approach, which can also be applied to the classical Hopfield network: we approximate the distribution of the crosstalk as Gaussian [26].

## B.2 Approach 2: Gaussian approximation

For the classical Hopfield network, the fact that the crosstalk can be expressed as a sum of $(P - 1)(N - 1)$ i.i.d. Rademacher random variables means that the classical central limit theorem implies that it tends in distribution to a Gaussian whenever $(P - 1)(N - 1)$ tends to infinity. By symmetry, the mean of the crosstalk is zero, while its variance is easily seen to be

$$\mathrm{var}(C) = \frac{P - 1}{N - 1}. \tag{B.21}$$

If we approximate the distribution of the crosstalk for $N$ and $P$ large but finite by a Gaussian, we therefore have

$$\mathbb{P}[C > 1] \approx H\left(\sqrt{\frac{N - 1}{P - 1}}\right) \tag{B.22}$$

where $H(x) = \mathrm{erfc}(x/\sqrt{2})/2$ is the Gaussian tail distribution function. We want to have $\mathbb{P}[C > 1] \to 0$, so we must have $(P - 1)/(N - 1) \to 0$. Then, we can use the asymptotic expansion [26]

$$H(\sqrt{z}) = \frac{1}{\sqrt{2\pi z}} \exp\left(-\frac{z}{2}\right) \left[1 + \mathcal{O}\left(\frac{1}{z}\right)\right] \quad \text{as} \quad z \to \infty \tag{B.23}$$

to obtain the heuristic Gaussian approximation

$$\mathbb{P}[C > 1] \approx \sqrt{\frac{(P - 1)}{2\pi(N - 1)}} \exp\left(-\frac{(N - 1)}{2(P - 1)}\right). \tag{B.24}$$

If we use this Gaussian approximation instead of the Hoeffding bound applied above, we can easily see that we will obtain identical estimates for the capacity with an error tolerance tending to zero. However, we have given up the rigor of the bound from Hoeffding's inequality, since we have not controlled the rate of convergence to the Gaussian tail probability. In particular, the Berry-Esseen theorem would give in this case a uniform additive error bound of $1/\sqrt{(P - 1)(N - 1)}$, which in the regime $P \sim N/[\alpha \log N]$ cannot compete with the factors of $N$ or $NP$ which we want $\mathbb{P}[C > 1]$ to overwhelm. We will not worry about this issue, as we are concerned more with whether we can get accurate capacity estimates that match numerical experiment than whether we can prove those estimates completely rigorously.

We can also use the Gaussian approximation to estimate the capacity for a non-zero error threshold $c$ at finite $N$. Concretely, if we demand that the union bound is saturated, i.e.,

$$NP\,\mathbb{P}[T_{HN}(\boldsymbol{\xi}^1)_1 \neq \xi_1^1] = c, \tag{B.25}$$

under the Gaussian approximation for the bitflip probability we have the self-consistent equation

$$NPH\left(\sqrt{\frac{N-1}{P-1}}\right) = c \tag{B.26}$$

for $P$, which we can re-write as

$$P - 1 = \frac{N-1}{[H^{-1}(c/NP)]^2}. \tag{B.27}$$

This is a transcendental self-consistent equation, which is not easy to solve analytically. However, we can make some progress at small $c/(NP)$. Using the asymptotic expansion of the inverse of the complementary error function [77], we have

$$[H^{-1}(x)]^2 = 2\operatorname{inverfc}(2x)^2 \tag{B.28}$$

$$\sim -\log\left[4\pi x^2 \log\left(\frac{1}{2x}\right)\right] \tag{B.29}$$

$$= -2\log(x) - \log(4\pi) - \log\log\left(\frac{1}{2x}\right) \tag{B.30}$$

$$\sim -2\log(x) \tag{B.31}$$

as $x \to 0$. Then, assuming $c$ is such that $-\log(c)$ is negligible relative to $\log(NP)$, we have

$$P \sim \frac{N}{2\log(NP)}, \tag{B.32}$$

which we can solve for $P$ as

$$P \sim \frac{N}{2W_0(N^2/2)}, \tag{B.33}$$

where $W_0$ is the principal branch of the Lambert-$W$ function [77]. But, at large $N$, we can use the asymptotic $W_0(N) \sim \log(N)$ to obtain the approximate scaling

$$P \sim \frac{N}{4\log(N)}, \tag{B.34}$$

which agrees with our earlier result. Conceptually, this intuition is consistent with there being a sharp transition in the thermodynamic limit, as proved rigorously by Bovier [34].

## C  DenseNet **Capacity**

In this Appendix, we analyze the capacity of the `DenseNet`. As introduced in Section 2.1 of the main text, there are two notions of robustness to consider: the robustness of a single transition and the robustness of the full sequence. For a fixed $N \in \{2, 3, \ldots\}$ and an error tolerance $c \in [0, 1)$, we define these two notions of capacity as

$$P_T(N, c) = \max\left\{P \in \{2, \ldots, 2^N\} : \mathbb{P}\left[\mathbf{T}_{DN}(\boldsymbol{\xi}^1) = \boldsymbol{\xi}^2\right] \geq 1 - c\right\} \tag{C.1}$$

and

$$P_S(N, c) = \max\left\{P \in \{2, \ldots, 2^N\} : \mathbb{P}\left[\cap_{\mu=1}^{P}\{\mathbf{T}_{DN}(\boldsymbol{\xi}^\mu) = \boldsymbol{\xi}^{\mu+1}\}\right] \geq 1 - c\right\}, \tag{C.2}$$

respectively.

Our goal is to approximately compute the capacity in the regime in which $N$ and $P$ are large. Following Petritis [33]'s approach to the HN, to make analytical progress, we can use a union bound to control the single-step error probability in terms of the probability of a single bitflip:

$$\mathbb{P}\left[\mathbf{T}_{DN}(\boldsymbol{\xi}^\mu) = \boldsymbol{\xi}^{\mu+1}\right]$$

$$= 1 - \mathbb{P}\left[\bigcup_{i=1}^{N}\{T_{DN}(\boldsymbol{\xi}^\mu)_i \neq \xi_i^{\mu+1}\}\right] \tag{C.3}$$

$$\geq 1 - \sum_{i=1}^{N}\mathbb{P}\left[T_{DN}(\boldsymbol{\xi}^\mu)_i \neq \xi_i^{\mu+1}\right] \tag{C.4}$$

$$= 1 - N\mathbb{P}[T_{DN}(\boldsymbol{\xi}^1)_1 \neq \xi_2^1]. \tag{C.5}$$

where we use the fact that all elements of all patterns are i.i.d. by assumption. We use a similar approach to control the sequence error probability in terms of the probability of a single bitflip:

$$\mathbb{P}\left[\bigcap_{\mu=1}^{P}\{\mathbf{T}_{DN}(\boldsymbol{\xi}^{\mu}) = \boldsymbol{\xi}^{\mu+1}\}\right]$$

$$= 1 - \mathbb{P}\left[\bigcup_{\mu=1}^{P}\bigcup_{i=1}^{N}\{T_{DN}(\boldsymbol{\xi}^{\mu})_i \neq \xi_i^{\mu+1}\}\right] \tag{C.6}$$

$$\geq 1 - \sum_{\mu=1}^{P}\sum_{i=1}^{N}\mathbb{P}\left[T_{DN}(\boldsymbol{\xi}^{\mu})_i \neq \xi_i^{\mu+1}\right] \tag{C.7}$$

$$= 1 - NP\mathbb{P}[T_{DN}(\boldsymbol{\xi}^1)_1 \neq \xi_2^1]. \tag{C.8}$$

Thus, as claimed in the main text, we have the lower bounds

$$P_T(N, c) \geq \max\left\{P \in \{2, \ldots, 2^N\} : NP\mathbb{P}[T_{DN}(\boldsymbol{\xi}^1)_1 \neq \xi_2^1] \leq c\right\} \tag{C.9}$$

and

$$P_S(N, c) \geq \max\left\{P \in \{2, \ldots, 2^N\} : NP\mathbb{P}[T_{DN}(\boldsymbol{\xi}^1)_1 \neq \xi_2^1] \leq c\right\}. \tag{C.10}$$

As introduced in the main text, for perfect recall, we want to take the threshold $c$ to be zero, or at least to tend to zero as $N$ and $P$ tend to infinity. The capacities estimated through this argument are lower bounds on the true capacities, as they are obtained from lower bounds on the true recall probability. However, we expect for these bounds to in fact be tight in the thermodynamic limit [33, 34].

By the definition of the `DenseNet` update rule with interaction function $f$ given in Equation (2), we have

$$T_{DN}(\boldsymbol{\xi}^1)_1 = \mathrm{sgn}\left[\sum_{\mu=1}^{P}\xi_1^{\mu+1}f\left(\frac{1}{N-1}\sum_{j=2}^{N}\xi_j^{\mu}\xi_j^1\right)\right] \tag{C.11}$$

and therefore the single-bitflip probability is

$$\mathbb{P}[T_{DN}(\boldsymbol{\xi}^1)_1 \neq \xi_1^2] = \mathbb{P}\left[\mathrm{sgn}\left[\sum_{\mu=1}^{P}\xi_1^{\mu+1}f\left(\frac{1}{N-1}\sum_{j=2}^{N}\xi_j^{\mu}\xi_j^1\right)\right] \neq \xi_1^2\right] \tag{C.12}$$

$$= \mathbb{P}\left[\xi_1^2\sum_{\mu=1}^{P}\xi_1^{\mu+1}f\left(\frac{1}{N-1}\sum_{j=2}^{N}\xi_j^{\mu}\xi_j^1\right) < 0\right] \tag{C.13}$$

$$= \mathbb{P}\left[f(1) + \xi_1^2\sum_{\mu=2}^{P}\xi_1^{\mu+1}f\left(\frac{1}{N-1}\sum_{j=2}^{N}\xi_j^{\mu}\xi_j^1\right) < 0\right] \tag{C.14}$$

For both the polynomial ($f(x) = x^d$) and exponential ($f(x) = e^{(N-1)(x-1)}$) interaction functions, $f(1) = 1$, and so

$$\mathbb{P}[T_{DN}(\boldsymbol{\xi}^1)_1 \neq \xi_1^2] = \mathbb{P}\left[\sum_{\mu=2}^{P}\xi_1^2\xi_1^{\mu+1}f\left(\frac{1}{N-1}\sum_{j=2}^{N}\xi_j^{\mu}\xi_j^1\right) < -1\right]. \tag{C.15}$$

We refer to the random variable

$$C = \sum_{\mu=2}^{P}\xi_1^2\xi_1^{\mu+1}f\left(\frac{1}{N-1}\sum_{j=2}^{N}\xi_j^{\mu}\xi_j^1\right) \tag{C.16}$$

on the left-hand-side of this inequality as the *crosstalk*, because it represents the effect of interference between the first pattern and all other patterns [26, 36].

We now observe that, as we have excluded self-interactions (i.e., the sum over neurons inside the interaction function does not include $j = 1$), we can use the periodic boundary conditions to shift indices as $\xi_1^\mu \leftarrow \xi_1^{\mu+1}$ for all $\mu$, yielding

$$C \stackrel{d}{=} \sum_{\mu=2}^{P} \xi_1^1 \xi_1^\mu f\left(\frac{1}{N-1}\sum_{j=2}^{N} \xi_j^\mu \xi_j^1\right) \tag{C.17}$$

Thus, the single-bitflip probability for this `DenseNet` is identical to that for the corresponding MHN with symmetric interactions. Then, we can use the fact that $\xi_j^\mu \xi_j^1 \stackrel{d}{=} \xi_j^\mu$ for all $\mu = 2, \dots, P$ to obtain

$$C \stackrel{d}{=} \sum_{\mu=2}^{P} \xi_1^\mu f\left(\frac{1}{N-1}\sum_{j=2}^{N} \xi_j^\mu\right). \tag{C.18}$$

Now, define the $P - 1$ random variables

$$\chi^\mu = \xi_1^\mu f\left(\frac{1}{N-1}\sum_{j=2}^{N} \xi_j^\mu\right) \tag{C.19}$$

for $\mu = 2, \dots, P$, such that the crosstalk is their sum,

$$C = \sum_{\mu=2}^{P} \chi^\mu. \tag{C.20}$$

As the patterns $\xi_j^\mu$ are i.i.d., $\chi^\mu$ are i.i.d. random variables of mean

$$\mathbb{E}[\chi^\mu] = \mathbb{E}[\xi_1^\mu]\mathbb{E}\left[f\left(\frac{1}{N-1}\sum_{j=2}^{N} \xi_j^\mu\right)\right] = 0 \tag{C.21}$$

and variance

$$\mathrm{var}(\chi^\mu) = \mathbb{E}\left[f\left(\frac{1}{N-1}\sum_{j=2}^{N} \xi_j^\mu\right)^2\right], \tag{C.22}$$

which is bounded from above for any sensible interaction function. We observe also that the distribution of each $\chi^\mu$ is symmetric because of the symmetry of the distribution of $\xi_1^\mu$. We will therefore simply write $\chi$ for any given $\chi^\mu$.

Then, the classical central limit theorem implies that the crosstalk tends in distribution to a Gaussian of mean zero and variance $(P-1)\,\mathrm{var}(\chi)$ as $P \to \infty$, at lease for any fixed $N$. However, we are interested in the joint limit in which $N, P \to \infty$ together. We will proceed by approximating the distribution of $C$ as Gaussian, and will not attempt to rigorously control its behavior in the joint limit.

Approximating the distribution of the crosstalk for $N, P \gg 1$ by a Gaussian, we then have

$$\mathbb{P}[T_{DN}(\boldsymbol{\xi}^1)_1 \neq \xi_1^2] \approx H\left(\frac{1}{\sqrt{(P-1)\,\mathrm{var}(\chi)}}\right) \tag{C.23}$$

where $H(x) = \mathrm{erfc}(x/\sqrt{2})/2$ is the Gaussian tail distribution function. We want to have $\mathbb{P}[T_{DN}(\boldsymbol{\xi}^1)_1 \neq \xi_1^2] \to 0$, so we must have $(P-1)\,\mathrm{var}(\chi) \to 0$. Then, we can use the asymptotic expansion [26]

$$H(\sqrt{z}) = \frac{1}{\sqrt{2\pi z}}\exp\left(-\frac{z}{2}\right)\left[1 + \mathcal{O}\left(\frac{1}{z}\right)\right] \quad \text{as} \quad z \to \infty \tag{C.24}$$

to obtain

$$\mathbb{P}[T_{DN}(\boldsymbol{\xi}^1)_1 \neq \xi_1^2] \approx \sqrt{\frac{(P-1)\,\mathrm{var}(\chi)}{2\pi}}\exp\left(-\frac{1}{2(P-1)\,\mathrm{var}(\chi)}\right). \tag{C.25}$$

For each model, we can evaluate $\mathrm{var}(\chi)$ and then determine the resulting predicted capacity.

As we did for the classic Hopfield network in Appendix B, we can estimate the capacity at finite $c$ within the Gaussian approximation by inverting the Gaussian tail distribution function. Concretely, under the union bound, we can estimate the transition capacity by solving

$$c = NH\left(\frac{1}{\sqrt{(P_T - 1)\,\mathrm{var}(\chi)}}\right), \tag{C.26}$$

which yields

$$P_T - 1 = \frac{1}{\mathrm{var}(\chi)[H^{-1}(c/N)]^2}, \tag{C.27}$$

and the sequence capacity by solving the transcendental self-consistent equation

$$c = NP_S H\left(\frac{1}{\sqrt{(P_S - 1)\,\mathrm{var}(\chi)}}\right), \tag{C.28}$$

which we can re-write as

$$P_S - 1 = \frac{1}{\mathrm{var}(\chi)[H^{-1}(c/NP_S)]^2}. \tag{C.29}$$

As in the classic Hopfield case, we can simplify these complicated equations somewhat by assuming that $c/N$ and $c/(NP_S)$ are small. Concretely, using the asymptotic

$$[H^{-1}(x)]^2 \sim -2\log(x) \tag{C.30}$$

for $x \to 0$, the transition capacity simplifies to

$$P_T - 1 \sim \frac{1}{2\,\mathrm{var}(\chi)\log(N)} \tag{C.31}$$

under the assumption that $-\log(c)$ is negligible relative to $\log(N)$. For the sequence capacity, we can follow an identical argument to that used for the classic Hopfield network to simplify the self-consistent equation to

$$P_S \sim \frac{1}{2\,\mathrm{var}(\chi)\log(NP_S)} \tag{C.32}$$

under the assumption that $-\log(c)$ is negligible relative to $\log(NP_S)$, which we can solve to obtain

$$P_S \sim \frac{1}{2\,\mathrm{var}(\chi)W_0[N/2\,\mathrm{var}(\chi)]}. \tag{C.33}$$

Assuming that $N/\mathrm{var}(\chi) \to \infty$ as $N \to \infty$, we can use the asymptotic $W_0(N) \sim \log(N)$ to obtain the asymptotic

$$P_S \sim \frac{1}{2\,\mathrm{var}(\chi)\log[N/\mathrm{var}(\chi)]}. \tag{C.34}$$

Our first check on the accuracy of the Gaussian approximation will be comparison of the resulting predictions for capacity with numerical experiment. As another diagnostic, we will consider the excess kurtosis $\varkappa = \kappa_4(C)/\kappa_2(C)$ for $\kappa_n(C)$ the $n$-th cumulant of $C$. If the distribution is indeed Gaussian, the excess kurtosis vanishes, while large values of the excess kurtosis indicate deviations from Gaussianity. By the additivity of cumulants, we have

$$\kappa_n(C) = (P-1)\kappa_n(\chi). \tag{C.35}$$

By symmetry, all odd cumulants of $\chi$—and therefore all odd cumulants of $C$—are identically zero. As noted above, we have

$$\mathrm{var}(\chi) = \kappa_2(\chi) = \mathbb{E}\left[f\left(\frac{1}{N-1}\sum_{j=2}^{N}\xi_j^\mu\right)^2\right]. \tag{C.36}$$

If $C$ is indeed Gaussian, then all cumulants above the second should vanish. As the third cumulant vanishes by symmetry, the leading possible correction to Gaussianity is the fourth cumulant, which as $\chi$ has zero mean is given by

$$\kappa_4(\chi) = \mathbb{E}[(\chi)^4] - 3\mathbb{E}[(\chi)^2]^2 \tag{C.37}$$

$$= \mathbb{E}\left[f\left(\frac{1}{N-1}\sum_{j=2}^{N}\xi_j^{\mu}\right)^4\right] - 3\mathbb{E}\left[f\left(\frac{1}{N-1}\sum_{j=2}^{N}\xi_j^{\mu}\right)^2\right]^2. \tag{C.38}$$

Rather than considering the fourth cumulant directly, we will consider the excess kurtosis

$$\varkappa = \frac{\kappa_4(C)}{\kappa_2(C)^2} = \frac{1}{P-1}\frac{\kappa_4(\chi)}{\kappa_2(\chi)^2}, \tag{C.39}$$

which is a more useful metric because it is normalized.

## C.1 Polynomial `DenseNet` Capacity

We first consider the Polynomial `DenseNet`, with interaction function $f(x) = x^d$ for $d \in \mathbb{N}_{>0}$. To compute the capacity, our goal is then to evaluate

$$\text{var}(\chi) = \mathbb{E}\left[\left(\frac{1}{N-1}\sum_{j=2}^{N}\xi_j^1\right)^{2d}\right] \tag{C.40}$$

at large $N$. From the central limit theorem, we expect

$$\mathbb{E}\left[\left(\frac{1}{N-1}\sum_{j=2}^{N}\xi_j^1\right)^{2d}\right] \sim \frac{(2d-1)!!}{(N-1)^d}. \tag{C.41}$$

We can make this quantitatively precise through the following straightforward argument. Let

$$\Xi = \frac{1}{\sqrt{N-1}}\sum_{j=2}^{N}\xi_j^2. \tag{C.42}$$

We then have immediately that the moment generating function of $\Xi$ is

$$M(t) = \mathbb{E}[e^{t\Xi}] = \cosh\left(\frac{t}{\sqrt{N-1}}\right)^{N-1}, \tag{C.43}$$

hence the cumulant generating function is

$$K(t) = \log M(t) = (N-1)\log\cosh\left(\frac{t}{\sqrt{N-1}}\right). \tag{C.44}$$

The function $x \mapsto \log\cosh(x)$ is an even function of $x$, and is analytic near the origin, with the first few orders of its MacLaurin series being

$$\log\cosh(x) = \frac{x^2}{2} - \frac{x^4}{12} + \mathcal{O}(x^6). \tag{C.45}$$

Then, the odd cumulants of $\Xi$ vanish—as we expect from symmetry—while the even cumulants obey

$$\kappa_{2k} = \frac{C_{2k}}{(N-1)^{k-1}} \tag{C.46}$$

for combinatorial factors $C_{2k}$ that do not scale with $N$. We have, in particular, $C_2 = 1$ and $C_4 = -2$. By the moments-cumulants formula, we have

$$\mathbb{E}[\Xi^{2k}] = B_{2k}(0, \kappa_2, 0, \kappa_4, \cdots, \kappa_{2k}) \tag{C.47}$$

for $B_{2k}$ the $2k$-th complete exponential Bell polynomial. From this, it follows that

$$\mathbb{E}[\Xi^{2k}] = (2k-1)!! + \mathcal{O}(N^{-1}), \tag{C.48}$$

as all cumulants other than $\kappa_2 = 1$ are $\mathcal{O}(N^{-1})$. Therefore, neglecting subleading terms, we have

$$\text{var}(\chi) = \mathbb{E}\left[\left(\frac{1}{N-1}\sum_{j=2}^{N}\xi_j^1\right)^{2d}\right] = \frac{(2d-1)!!}{N^d}\left[1 + \mathcal{O}\left(\frac{1}{N}\right)\right]. \tag{C.49}$$

Following the general arguments above, we then approximate

$$\mathbb{P}[T_{DN}(\boldsymbol{\xi}^1)_1 \neq \xi_1^2] \sim \sqrt{\frac{P(2d-1)!!}{2\pi N^d}}\exp\left(-\frac{N^d}{2P(2d-1)!!}\right). \tag{C.50}$$

To determine the single-transition capacity following the argument in Section 2.1, we must determine how large we can take $P = P(N)$ such that $N\mathbb{P}[T_{DN}(\boldsymbol{\xi}^1)_1 \neq \xi_1^2] \to 0$. Following the requirement that $P\text{var}(\chi) \to 0$, we make the *Ansatz*

$$P \sim \frac{N^d}{\alpha(2d-1)!!\,\log N} \tag{C.51}$$

for some $\alpha$. We then have

$$N\mathbb{P}[T_{DN}(\boldsymbol{\xi}^1)_1 \neq \xi_1^2] \sim \sqrt{\frac{1}{2\pi\alpha\log N}}N^{1-\alpha/2}. \tag{C.52}$$

This tends to zero if $\alpha \geq 2$, meaning that the predicted capacity in this case is

$$P_T \sim \frac{N^d}{2(2d-1)!!\,\log N}. \tag{C.53}$$

We now want to determine the sequence capacity, which requires the stronger condition $NP\mathbb{P}[T_{DN}(\boldsymbol{\xi}^1)_1 \neq \xi_1^2] \to 0$. Again making the *Ansatz*

$$P \sim \frac{N^d}{\alpha(2d-1)!!\,\log N} \tag{C.54}$$

for some $\alpha$, we then have

$$NP\mathbb{P}[T_{DN}(\boldsymbol{\xi}^1)_1 \neq \xi_1^2] \sim \frac{1}{\sqrt{2\pi}(2d-1)!!\,(\alpha\log N)^{3/2}}N^{d+1-\alpha/2}, \tag{C.55}$$

which tends to zero if $\alpha \geq 2d+2$. Then, the predicted sequence capacity is

$$P_S \sim \frac{N^d}{2(d+1)(2d-1)!!\,\log N}. \tag{C.56}$$

If we consider the alternative asymptotic formulas obtained above from the finite-$c$ argument, we have

$$P_T \sim \frac{1}{2\,\text{var}(\chi)\log(N)} \sim \frac{N^d}{2(2d-1)!!\,\log(N)} \tag{C.57}$$

and

$$P_S \sim \frac{1}{2\,\text{var}(\chi)\log[N/\text{var}(\chi)]} \sim \frac{N^d}{2(2d-1)!!\,\log[N^{d+1}/(2d-1)!!]} \sim \frac{N^d}{2(d+1)(2d-1)!!\,\log(N)}, \tag{C.58}$$

which agree with these results. For evidence of the finite-$c$ argument for the polynomial `DenseNet`, observe Figure C.1.

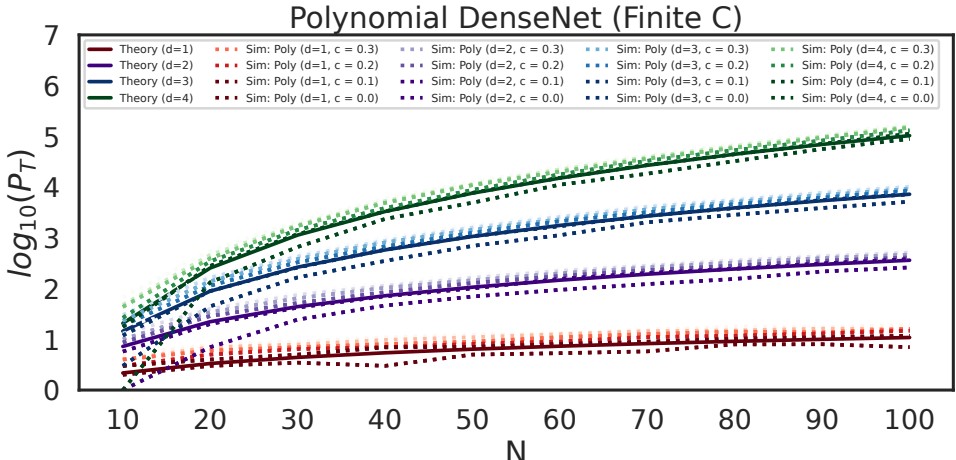

Figure C.1: The transition capacity of the polynomial `DenseNet` is demonstrated for different values of error tolerance $c$. We see that even for $c \neq 0$, we get similar scaling curves although the capacities slightly increase consistently as we increase $c$, indicated by a transition from dark to light. We plot from $c = 0.0$ to $c = 0.5$ for each degree $d$, with the legend labeling curves up to $c = 0.3$ to demonstrate the general trend.

Using the Gaussian approximation for moments of $\chi$ given above, we can easily work out that

$$\kappa_4(\chi) = \mathbb{E}[(\chi)^4] - 3\mathbb{E}[(\chi)^2] \tag{C.59}$$

$$= \mathbb{E}\left[f\left(\frac{1}{N-1}\sum_{j=2}^{N}\xi_j^{\mu}\right)^4\right] - 3\mathbb{E}\left[f\left(\frac{1}{N-1}\sum_{j=2}^{N}\xi_j^{\mu}\right)^2\right]^2 \tag{C.60}$$

$$= \frac{1}{N^{2d}}\{(4d-1)!! - 3[(2d-1)!!]^2\}\left[1 + \mathcal{O}\left(\frac{1}{N}\right)\right]. \tag{C.61}$$

Then, the excess kurtosis of the Polynomial `DenseNet`'s crosstalk is

$$\varkappa = \frac{1}{P-1}\left[\frac{(4d-1)!!}{[(2d-1)!!]^2} - 3\right]\left[1 + \mathcal{O}\left(\frac{1}{N}\right)\right]. \tag{C.62}$$

Thus, for the Polynomial `DenseNet`, we expect the excess kurtosis to be small for any fixed $d$ so long as $P$ and $N$ are both fairly large, without any particular requirement on their relationship. In particular, under the Gaussian approximation we predicted above that the transition and sequence capacities should both scale as

$$P \sim \frac{N^d}{\alpha_d \log N}, \tag{C.63}$$

where $\alpha_d$ depends on $d$ but not on $N$. This gives an excess kurtosis of

$$\varkappa = \frac{\alpha_d \log N}{N^d}\left[\frac{(4d-1)!!}{[(2d-1)!!]^2} - 3\right]\left[1 + \mathcal{O}\left(\frac{1}{N}\right)\right] \tag{C.64}$$

which for any fixed $d$ rapidly tends to zero with increasing $N$. This suggests that the Gaussian approximation should be reasonably accurate even at modest $N$, but of course does not constitute a proof of its accuracy because we have not considered higher cumulants. However, this matches the results of numerical simulations shown in Figure 2.

## C.2 Exponential `DenseNet` capacity

We now turn our attention to the Exponential `DenseNet`, with separation function $f(x) = e^{(N-1)(x-1)}$. In this case, we have

$$\text{var}(\chi) = \exp[-2(N-1)]\mathbb{E}\left[\exp\left(2\sum_{j=2}^{N}\xi_j^2\right)\right] \tag{C.65}$$

$$= \exp[-2(N-1)]\prod_{j=2}^{N}\mathbb{E}\left[\exp\left(2\xi_j^2\right)\right] \tag{C.66}$$

$$= \exp[-2(N-1)]\cosh(2)^{N-1} \tag{C.67}$$

$$= \frac{1}{\beta^{N-1}}, \tag{C.68}$$

where we have defined the constant

$$\beta = \frac{\exp(2)}{\cosh(2)} \simeq 1.96403. \tag{C.69}$$

Then, we have the Gaussian approximation

$$\mathbb{P}[T_{DN}(\boldsymbol{\xi}^1)_1 \neq \xi_1^2] \sim \sqrt{\frac{P}{2\pi\beta^{N-1}}}\exp\left(-\frac{\beta^{N-1}}{2P}\right). \tag{C.70}$$

As in the polynomial case, we first determine the single-transition capacity by demanding that $N\mathbb{P}[T_{DN}(\boldsymbol{\xi}^1)_1 \neq \xi_1^2] \to 0$. We plug in the *Ansatz*

$$P \sim \frac{\beta^{N-1}}{\alpha\log N} \tag{C.71}$$

for some $\alpha$, which yields

$$N\mathbb{P}[T_{DN}(\boldsymbol{\xi}^1)_1 \neq \xi_1^2] \sim \sqrt{\frac{1}{2\pi\alpha\log N}}N^{1-\alpha/2}. \tag{C.72}$$

This tends to zero if $\alpha \geq 2$, which gives a predicted capacity of

$$P_T \sim \frac{\beta^{N-1}}{2\log N}. \tag{C.73}$$

Considering the sequence capacity, which again requires that $NP\mathbb{P}[T_{DN}(\boldsymbol{\xi}^1)_1 \neq \xi_1^2] \to 0$, we plug in the *Ansatz*

$$P \sim \frac{\beta^{N-1}}{\alpha N}, \tag{C.74}$$

which yields

$$NP\mathbb{P}[T_{DN}(\boldsymbol{\xi}^1)_1 \neq \xi_1^2] \sim \frac{1}{\alpha\beta}\sqrt{\frac{1}{2\pi\alpha N}}\exp\left[\left(\log\beta - \frac{\alpha}{2}\right)N\right]. \tag{C.75}$$

This tends to zero for $\alpha \geq 2\log\beta$, meaning that the predicted capacity is in this case

$$P_S \sim \frac{\beta^{N-1}}{2\log(\beta)N}. \tag{C.76}$$

Therefore, while the ratio of the predicted single-transition to sequence capacities is finite for the Polynomial `DenseNet`—it is simply $P_S/P_T \sim d+1$—for the Exponential `DenseNet` it tends to zero as $P_S/P_T \sim \log N/[\log(\beta)N]$.

Using the asymptotic formulas obtained above from the finite-$c$ argument, we have

$$P_T \sim \frac{1}{2\,\text{var}(\chi)\log(N)} = \frac{\beta^{N-1}}{2\log(N)} \tag{C.77}$$

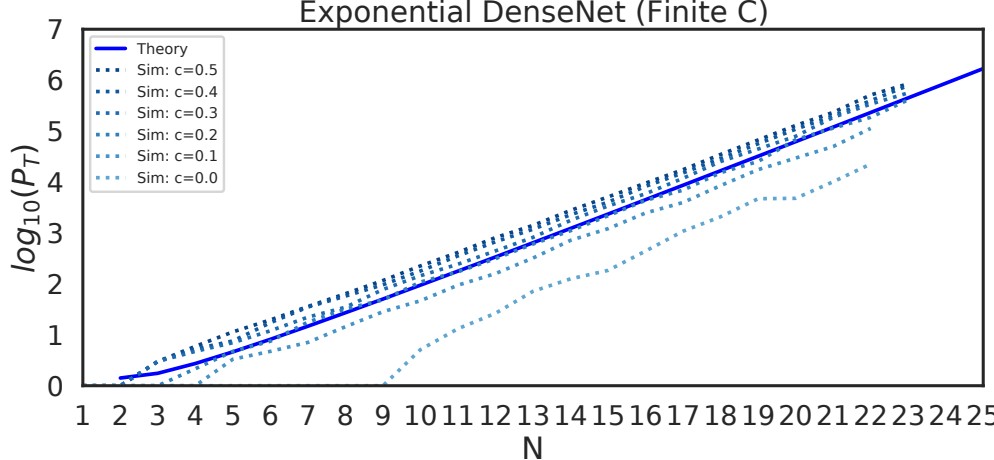

Figure C.2: The transition capacity of the exponential `DenseNet` is demonstrated for different values of error tolerance $c$. We see that even for $c \neq 0$, we get similar scaling curves although the capacities slightly increase consistently as we increase $c$.

and

$$P_S \sim \frac{1}{2\operatorname{var}(\chi)\log[N/\operatorname{var}(\chi)]} = \frac{\beta^{N-1}}{2\log[N\beta^{N-1}]} \sim \frac{\beta^{N-1}}{2\log(\beta)N}, \tag{C.78}$$

which agree with these results. For evidence of the finite-$c$ argument for the exponential `DenseNet`, observe Figure C.2.

Now considering the fourth cumulant, we can easily compute

$$\kappa_4(\chi) = \left(\frac{\cosh(4)}{\exp(4)}\right)^{N-1} - 3\left(\frac{\cosh(2)^2}{\exp(4)}\right)^{N-1}, \tag{C.79}$$

which yields an excess kurtosis of

$$\varkappa = \frac{1}{P-1}\left[\left(\frac{\cosh(4)}{\cosh(2)^2}\right)^{N-1} - 3\right]. \tag{C.80}$$

For this to be small, $P$ must be exponentially large in $N$, which contrasts with the situation for the Polynomial `DenseNet`, in which the excess kurtosis is small for any reasonably large $P$. If we consider taking

$$P \sim \frac{\beta^{N-1}}{\alpha\log N}, \tag{C.81}$$

for a constant $\alpha$, as the Gaussian theory predicts for the Exponential `DenseNet` transition capacity, we have

$$\varkappa \sim \frac{\alpha\log N}{\beta^{N-1}}\left[\left(\frac{\cosh(4)}{\cosh(2)^2}\right)^{N-1} - 3\right] \tag{C.82}$$

$$\sim \alpha\log N\left(\frac{\cosh(4)}{\exp(2)\cosh(2)}\right)^{N-1} \tag{C.83}$$

$$\simeq \alpha\log(N)(0.9823)^{N-1}. \tag{C.84}$$

This tends to zero as $N$ increases, but only very slowly. In particular, $\log(N)(0.9823)^{N-1}$ increases with $N$ up to around $N \simeq 19$, where it attains a maximum value around 2, before decreasing towards zero. The situation is even worse for the sequence capacity, for which the Gaussian theory predicts

$$P \sim \frac{\beta^{N-1}}{\alpha N}, \tag{C.85}$$

yielding

$$\varkappa \sim \frac{\alpha N}{\beta^{N-1}} \left[ \left( \frac{\cosh(4)}{\cosh(2)^2} \right)^{N-1} - 3 \right] \tag{C.86}$$

$$\sim \alpha N \left( \frac{\cosh(4)}{\exp(2)\cosh(2)} \right)^{N-1} \tag{C.87}$$

$$\simeq \alpha N (0.9823)^{N-1}. \tag{C.88}$$

$N(0.9823)^{N-1}$ increases with $N$ up to around $N \simeq 56$, where it attains a value of approximately 21.

Taken together, these results suggest that we might expect substantial finite-size corrections to the Gaussian theory's prediction for the capacity. In particular, as the excess kurtosis of the crosstalk is positive, the tails of the crosstalk distribution should be heavier-than-Gaussian, suggesting that the Gaussian theory should overestimate the true capacity. This holds provided that the lower bound on the memorization probability resulting from the union bound is reasonably tight.

## D  Bounding the polynomial `DenseNet` capacity

Here, we adapt Demircigil et al. [31]'s proof of a rigorous asymptotic lower bound on the polynomial MHN's capacity to obtain a rigorous asymptotic lower bound on the DenseNet capacity. This proof is a step-by-step adaptation of the proof of Theorem 1.2 of Demircigil et al. [31], which we spell out in detail for clarity.

Our objective is to obtain an upper bound on the single-bitflip probability

$$\mathbb{P}[T_{DN}(\boldsymbol{\xi}^1)_1 \neq \xi_2^1] \tag{D.1}$$

which we have argued can be expressed in terms of the crosstalk $C$ as

$$\mathbb{P}[T_{DN}(\boldsymbol{\xi}^1)_1 \neq \xi_2^1] = \mathbb{P}[C < -1] \tag{D.2}$$

for

$$C \stackrel{d}{=} \sum_{\mu=2}^{P} \xi_1^\mu \left( \frac{1}{N-1} \sum_{j=2}^{N} \xi_j^\mu \right)^d. \tag{D.3}$$

Our goal is to prove the following: First, letting $\alpha > 2(2d-1)!!$ and $P = N^d/(\alpha \log N)$, we have

$$N\mathbb{P}[T_{DN}(\boldsymbol{\xi}^1)_1 \neq \xi_2^1] \to 0 \tag{D.4}$$

as $N \to \infty$. Second, letting $\alpha > 2(d+1)(2d-1)!!$ and $P = N^d/(\alpha \log N)$, we have

$$NP\mathbb{P}[T_{DN}(\boldsymbol{\xi}^1)_1 \neq \xi_2^1] \to 0 \tag{D.5}$$

as $N \to \infty$.

By Chernoff's inequality (also known as the exponential Chebyschev inequality) [76], we then have

$$\mathbb{P}[T_{DN}(\boldsymbol{\xi}^1)_1 \neq \xi_2^1] = \mathbb{P}\left[ \sum_{\mu=2}^{P} \xi_1^\mu \left( \sum_{j=2}^{N} \xi_j^\mu \right)^d < -(N-1)^d \right] \tag{D.6}$$

$$\leq e^{-t(N-1)^d} \mathbb{E} \exp\left[ -t \sum_{\mu=2}^{P} \xi_1^\mu \left( \sum_{j=2}^{N} \xi_j^\mu \right)^d \right] \tag{D.7}$$

for any $t > 0$. Using the fact that the pattern elements are i.i.d., we have

$$\mathbb{E} \exp\left[ -t \sum_{\mu=2}^{P} \xi_1^\mu \left( \sum_{j=2}^{N} \xi_j^\mu \right)^d \right] = \left\{ \mathbb{E} \exp\left[ -t\xi_1^\mu \left( \sum_{j=2}^{N} \xi_j^\mu \right)^d \right] \right\}^{P-1} \tag{D.8}$$

$$= \left\{ \mathbb{E} \cosh\left[ t \left( \sum_{j=2}^{N} \xi_j^\mu \right)^d \right] \right\}^{P-1}. \tag{D.9}$$

Now, let

$$M = \frac{1}{\sqrt{N-1}} \sum_{j=2}^{N} \xi_j^\mu, \tag{D.10}$$

and expand the expectation as a sum over the possible values $m \in \{0, \pm(N-1)^{-1/2}, \ldots, \pm(N-1)^{1/2}\}$ of $M$:

$$\mathbb{E} \cosh \left[ t \left( \sum_{j=2}^{N} \xi_j^\mu \right)^d \right] = \sum_m \cosh \left[ t(N-1)^{d/2} m^d \right] \mathbb{P}[M = m]. \tag{D.11}$$

For $N \gg 1$, the distribution of $M$ is nearly Gaussian. We thus split the sum over $m$ to allow us to treat tail events separately. We fix $\beta > 0$, and split the sum at $\log(N)^\beta$:

$$\sum_m \cosh \left[ t(N-1)^{d/2} m^d \right] \mathbb{P}[M = m] = \sum_{|m| \leq \log(N)^\beta} \cosh \left[ t(N-1)^{d/2} m^d \right] \mathbb{P}[M = m]$$

$$+ \sum_{\log(N)^\beta < |m| \leq \sqrt{N}} \cosh \left[ t(N-1)^{d/2} m^d \right] \mathbb{P}[M = m], \tag{D.12}$$

where we have used the fact that $M \leq \sqrt{N-1}$.

We first consider the tail sum over $|m| > \log(N)^\beta$. As cosh is even and non-decreasing in the modulus of its argument, we have

$$\sum_{\log(N)^\beta < |m| \leq \sqrt{N}} \cosh \left[ t(N-1)^{d/2} m^d \right] \mathbb{P}[M = m] \tag{D.13}$$

$$\leq 2 \cosh \left[ t(N-1)^d \right] \mathbb{P}[M > \log(N)^\beta] \tag{D.14}$$

$$\leq 2 \cosh \left[ t(N-1)^d \right] \exp \left( -\frac{1}{2} \log(N)^{2\beta} \right) \tag{D.15}$$

$$\leq 2 \exp \left[ t(N-1)^d - \frac{1}{2} \log(N)^{2\beta} \right], \tag{D.16}$$

where in the second line we have applied Hoeffding's inequality to bound $\mathbb{P}[M > \log(N)^\beta]$ from above, and in the third line we have used the bound $\cosh(z) \leq \exp(z)$ for any $z > 0$.

We now consider the sum over $|m| \leq \log(N)^\beta$. Using the bound $\cosh(z) \leq \exp(z^2/2)$, we have

$$\sum_{|m| \leq \log(N)^\beta} \cosh \left[ t(N-1)^{d/2} m^d \right] \mathbb{P}[M = m] \leq \sum_{|m| \leq \log(N)^\beta} \exp \left[ \frac{1}{2} t^2 (N-1)^d m^{2d} \right] \mathbb{P}[M = m]. \tag{D.17}$$

Using the series expansion of the exponential, we have

$$\sum_{|m| \leq \log(N)^\beta} \exp \left[ \frac{1}{2} t^2 (N-1)^d m^{2d} \right] \mathbb{P}[M = m] \tag{D.18}$$

$$= \sum_{|m| \leq \log(N)^\beta} \left\{ 1 + \frac{1}{2} t^2 (N-1)^d m^{2d} + \sum_{k=2}^{\infty} \frac{(t^2 (N-1)^d m^{2d})^k}{2^k k!} \right\} \mathbb{P}[M = m] \tag{D.19}$$

$$= \mathbb{P}[|M| \leq \log(N)^\beta] + \frac{1}{2} t^2 (N-1)^d \sum_{|m| \leq \log(N)^\beta} m^{2d} \mathbb{P}[M = m]$$

$$+ \sum_{|m| \leq \log(N)^\beta} \left\{ \sum_{k=2}^{\infty} \frac{(t^2 (N-1)^d m^{2d})^k}{2^k k!} \right\} \mathbb{P}[M = m] \tag{D.20}$$

where on the third line we have used the linearity of summation. We will now bound each of the three contributions. The first term is trivially bounded from above by 1:

$$\mathbb{P}[|M| \leq \log(N)^\beta] \leq 1. \tag{D.21}$$

To handle the second, we first observe that

$$\sum_{|m| \leq \log(N)^\beta} m^{2d} \mathbb{P}[M = m] \leq \mathbb{E}[m^{2d}]. \tag{D.22}$$

Then, we observe that as $m$ is the normalized sum of $N - 1$ Rademacher random variables, its moments tend to those of a standard normal from below as $N \to \infty$, and are for any $N$ strictly bounded from above by those of the standard normal. Therefore, we have

$$\mathbb{E}[m^{2d}] \leq (2d - 1)!!. \tag{D.23}$$

To handle the third term, we first use the fact that for any $|m| \leq \log(N)^\beta$ we have $m^{2d} \leq \log(N)^{2\beta d}$, which gives

$$\sum_{|m| \leq \log(N)^\beta} \left\{ \sum_{k=2}^\infty \frac{(t^2(N-1)^d m^{2d})^k}{2^k k!} \right\} \mathbb{P}[M = m]$$

$$\leq \sum_{|m| \leq \log(N)^\beta} \left\{ \sum_{k=2}^\infty \frac{(t^2(N-1)^d \log(N)^{2\beta d})^k}{2^k k!} \right\} \mathbb{P}[M = m] \tag{D.24}$$

$$\leq \mathbb{P}[|M| \leq \log(N)^\beta] \sum_{k=2}^\infty \frac{(t^2(N-1)^d \log(N)^{2\beta d})^k}{2^k k!} \tag{D.25}$$

$$\leq \sum_{k=2}^\infty \frac{(t^2(N-1)^d \log(N)^{2\beta d})^k}{2^k k!} \tag{D.26}$$

At this point, [31] uses the bound

$$\sum_{k=2}^\infty \frac{(t^2(N-1)^d \log(N)^{2\beta d})^k}{2^k k!} \leq \frac{1}{4}(e-2)(t^2(N-1)^d \log(N)^{2\beta d})^2 \tag{D.27}$$

which holds provided that we choose the arbitrary parameter $t$ such that

$$t^2(N-1)^d \log(N)^{2\beta d} \leq 2. \tag{D.28}$$

Assuming that condition is satisfied, we can then combine these results to obtain

$$\sum_{|m| \leq \log(N)^\beta} \cosh\left[ t(N-1)^{d/2} m^d \right] \mathbb{P}[M = m] \tag{D.29}$$

$$\leq 1 + \frac{1}{2} t^2(N-1)^d (2d-1)!! + \frac{1}{4}(e-2)(t^2(N-1)^d \log(N)^{2\beta d})^2 \tag{D.30}$$

$$\leq 1 + \frac{1}{2} t^2(N-1)^d (2d-1)!! + \frac{1}{4}(t^2(N-1)^d \log(N)^{2\beta d})^2 \tag{D.31}$$

$$\leq \exp\left[ \frac{1}{2} t^2(N-1)^d (2d-1)!! + \frac{1}{4}(t^2(N-1)^d \log(N)^{2\beta d})^2 \right], \tag{D.32}$$

where on the the second line we have used the fact that $e - 2 \simeq 0.718\ldots < 1$ and on the third line we have used the bound $1 + x \leq \exp(x)$ for $x \geq 0$.

Combining this result with the bound on the tail sum obtained previously, we have that

$$\mathbb{E}\cosh\left[ t \left( \sum_{j=2}^N \xi_j^\mu \right)^d \right] \leq \exp\left[ \frac{1}{2} t^2(N-1)^d (2d-1)!! + \frac{1}{4}(t^2(N-1)^d \log(N)^{2\beta d})^2 \right]$$

$$+ 2\exp\left[ t(N-1)^d - \frac{1}{2}\log(N)^{2\beta} \right] \tag{D.33}$$

for any $\beta > 1/2$ and

$$0 < t \leq \sqrt{\frac{2}{(N-1)^d \log(N)^{2\beta d}}}.$$ (D.34)

Therefore, we have

$$\mathbb{P}[T_{DN}(\boldsymbol{\xi}^1)_1 \neq \xi_2^1] \leq e^{-t(N-1)^d} \left\{ \exp\left[\frac{1}{2}t^2(N-1)^d(2d-1)!! + \frac{1}{4}(t^2(N-1)^d \log(N)^{2\beta d})^2\right]\right.$$

$$\left. + 2\exp\left[t(N-1)^d - \frac{1}{2}\log(N)^{2\beta}\right]\right\}^{P-1}$$ (D.35)

subject to these conditions on $\beta$ and $t$.

We now want to determine the single-transition and full-sequence capacities. To do so, we fix $\alpha > 0$, and let $P = N^d/(\alpha \log N)$. As $t$ is arbitrary, fix $\gamma > 0$, and let $t = \gamma/P$. For our choice of $P$, this gives

$$t^2(N-1)^d \log(N)^{2\beta d} = \gamma^2 \alpha^2 \frac{(N-1)^d}{N^{2d}} \log(N)^{2(\beta d+1)}$$ (D.36)

which is clearly less than 2 for $N$ sufficiently large. Therefore, we can apply the bound obtained above, which for this choice of $t$ simplifies to

$$\mathbb{P}[T_{DN}(\boldsymbol{\xi}^1)_1 \neq \xi_2^1]$$

$$\leq e^{-t(N-1)^d} \left\{ \exp\left[\frac{1}{2}\gamma^2 \alpha^2(2d-1)!!\frac{\log(N)^2}{N^d} + \frac{1}{4}\gamma^4 \alpha^4 \frac{\log(N)^{4(\beta d-1)}}{N^{2d}}\right][1 + o(1)]\right.$$

$$\left. + 2\exp\left[\left(\gamma\alpha - \frac{1}{2}\log(N)^{2\beta-1}\right)\log(N)\right][1 + o(1)]\right\}^{P-1}.$$ (D.37)

We can see that the first term in the curly braces tends to 1 with increasing $N$—as its exponent tends to zero—while the second term tends to zero as the term in the round brackets within the exponent is negative for sufficiently large $N$ provided that $\beta > 1/2$. We may therefore neglect the second term, which gives the simplification

$$\mathbb{P}[T_{DN}(\boldsymbol{\xi}^1)_1 \neq \xi_2^1] \leq \exp\left[-\alpha\gamma\left(1 - \frac{1}{2}\gamma(2d-1)!!\right)\log(N)\right][1 + o(1)].$$ (D.38)

To determine the single-transition capacity under the union bound, we want $N\mathbb{P}[T_{DN}(\boldsymbol{\xi}^1)_1 \neq \xi_2^1]$ to tend to zero. We have

$$N\mathbb{P}[T_{DN}(\boldsymbol{\xi}^1)_1 \neq \xi_2^1] \leq \exp\left\{\left[1 - \alpha\gamma\left(1 - \frac{1}{2}\gamma(2d-1)!!\right)\right]\log(N)\right\}[1 + o(1)].$$ (D.39)

For this bound to tend to zero, we should have

$$1 - \alpha\gamma\left(1 - \frac{1}{2}\gamma(2d-1)!!\right) < 0.$$ (D.40)

As $\gamma$ is arbitrary, we may let $\gamma = 1/(2d-1)!!$, hence the required condition is clearly satisfied if

$$\alpha > 2(2d-1)!!,$$ (D.41)

as predicted by the Gaussian approximation. Next, to determine the sequence capacity, we want $NP\mathbb{P}[T_{DN}(\boldsymbol{\xi}^1)_1 \neq \xi_2^1]$ to tend to zero. We have

$$NP\mathbb{P}[T_{DN}(\boldsymbol{\xi}^1)_1 \neq \xi_2^1] \leq \frac{1}{\alpha \log N} \exp\left\{\left[d + 1 - \alpha\gamma\left(1 - \frac{1}{2}\gamma(2d-1)!!\right)\right]\log(N)\right\}[1 + o(1)],$$ (D.42)

hence an identical line of reasoning to that used for the single-transition capacity shows that we must have

$$\alpha \geq 2(d+1)(2d-1)!!.$$ (D.43)

Again, this agrees with the Gaussian theory.

## E   Generalized pseudoinverse rule capacity

Here, we show that the generalized pseudoinverse rule can perfectly recall any sequence of linearly-independent patterns. We recall from (11) that the GPI update rule is

$$T_{GPI}(\mathbf{S})_i = \text{sgn}\left[\sum_{\mu=1}^{P}\xi_i^{\mu+1}f\left(\sum_{\nu=1}^{P}(O^+)^{\mu\nu}m^{\nu}(\mathbf{S})\right)\right] \tag{E.1}$$

for

$$O^{\mu\nu} = \frac{1}{N}\sum_{j=1}^{N}\xi_j^{\mu}\xi_j^{\nu} \tag{E.2}$$

the Gram matrix of the patterns. If the patterns are linearly independent, then $\mathbf{O}$ is full rank, and the pseudoinverse reduces to the ordinary inverse: $\mathbf{O}^+ = \mathbf{O}^{-1}$. Under this assumption, we have

$$T_{GPI}(\boldsymbol{\xi}^{\mu})_i = \text{sgn}\left[\sum_{\nu=1}^{P}\xi_i^{\nu+1}f(\delta^{\mu\nu})\right] \tag{E.3}$$

$$= \text{sgn}\left[f(1)\xi_i^{\mu+1} + f(0)\sum_{\nu\neq\mu}\xi_i^{\nu+1}\right], \tag{E.4}$$

for all $\mu$ and $i$, hence for separation functions satisfying $f(1) > 0$ and $|f(0)| < f(1)/(P-1)$ we are guaranteed to have $T_{GPI}(\boldsymbol{\xi}^{\mu})_i = \xi_i^{\mu+1}$ as desired. For $f(x) = x^d$, this condition is always satisfied as $f(0) = 0$ and $f(1) = 1$. For $f(x) = e^{(N-1)(x-1)}$, we have $f(0) = e^{-(N-1)}$ and $f(1) = 1$; the condition $P - 1 < e^{N-1}$ must therefore be satisfied. However, as $P \leq N$ is required for linear independence, this condition is satisfied so long as $N > 3$.

## F   `MixedNet` Capacity

In this Appendix, we compute the capacity of the mixed network, which from the update rule defined in (14) has

$$T_{MN}(\boldsymbol{\xi}^1)_1 = \text{sgn}\left\{\sum_{\mu=1}^{P}\left[\xi_1^{\mu}f_S\left(\frac{1}{N-1}\sum_{j=2}^{N}\xi_j^{\mu}\xi_j^1\right) + \lambda\xi_1^{\mu+1}f_A\left(\frac{1}{N-1}\sum_{j=2}^{N}\xi_j^{\mu}\xi_j^1\right)\right]\right\}. \tag{F.1}$$

Then, assuming that $f_S(1) = f_A(1) = 1$ as is true for the interaction functions considered here, we have

$$\mathbb{P}[T_{MN}(\boldsymbol{\xi}^1)_1 \neq \xi_1^2] \tag{F.2}$$

$$= \mathbb{P}\left\{\xi_1^2\left[\sum_{\mu=1}^{P}\xi_1^{\mu}f_S\left(\frac{1}{N-1}\sum_{j=2}^{N}\xi_j^{\mu}\xi_j^1\right) + \lambda\sum_{\mu=1}^{P}\xi_1^{\mu+1}f_A\left(\frac{1}{N-1}\sum_{j=2}^{N}\xi_j^{\mu}\xi_j^1\right)\right] < 0\right\} \tag{F.3}$$

$$= \mathbb{P}\left\{C < -\lambda\right\}, \tag{F.4}$$

where we have defined the crosstalk

$$C = \xi_1^2\xi_1^1 + \sum_{\mu=2}^{P}\xi_1^2\xi_1^{\mu}f_S\left(\frac{1}{N-1}\sum_{j=2}^{N}\xi_j^{\mu}\xi_j^1\right) + \lambda\sum_{\mu=2}^{P}\xi_1^2\xi_1^{\mu+1}f_A\left(\frac{1}{N-1}\sum_{j=2}^{N}\xi_j^{\mu}\xi_j^1\right). \tag{F.5}$$

For $j = 2, \ldots, N$ and $\mu = 2, \ldots, P$, we have the equality in distribution $\xi_j^{\mu}\xi_j^1 \overset{d}{=} \xi_j^{\mu}$, hence

$$C \overset{d}{=} \xi_1^2\xi_1^1 + \sum_{\mu=2}^{P}\xi_1^2\xi_1^{\mu}f_S(\Xi^{\mu}) + \lambda\sum_{\mu=2}^{P}\xi_1^2\xi_1^{\mu+1}f_A(\Xi^{\mu}). \tag{F.6}$$

where to lighten our notation we define

$$\Xi^\mu = \frac{1}{N-1}\sum_{j=2}^{N}\xi_j^\mu. \tag{F.7}$$

However, unlike in the `DenseNet`, we cannot similarly simplify the terms outside the separation functions. Recalling that we have assumed periodic boundary conditions, we have

$$C = \xi_1^2\xi_1^1 + \lambda\xi_1^2\xi_1^1 f_A(\Xi^P) + f_S(\Xi^2) + \sum_{\mu=3}^{P}\xi_1^2\xi_1^\mu f_S(\Xi^\mu) + \lambda\sum_{\mu=2}^{P-1}\xi_1^2\xi_1^{\mu+1}f_A(\Xi^\mu) \tag{F.8}$$

$$\overset{d}{=} \xi_1^1 + C_1 + C_2 + C_3 + C_4, \tag{F.9}$$

where we have defined

$$C_1 = f_S(\Xi^2) + \lambda\,\xi_1^3 f_A(\Xi^2), \tag{F.10}$$

$$C_2 = \xi_1^P f_S(\Xi^P) + \lambda\xi_1^1 f_A(\Xi^P), \tag{F.11}$$

$$C_3 = \sum_{\mu=3}^{P-1}\xi_1^\mu f_S(\Xi^\mu), \quad \text{and} \tag{F.12}$$

$$C_4 = \lambda\sum_{\mu=3}^{P-1}\xi_1^{\mu+1}f_A(\Xi^\mu). \tag{F.13}$$

Importantly, in this case the influence of $\xi_1^1$ on the crosstalk is $\mathcal{O}(1)$, and the distribution is not well-approximated by a single Gaussian. Instead, as shown in Figure F.1, it is bimodal. We will therefore approximate it by a mixture of two Gaussians, one for each value of $\xi_1^1$. This approximation can be justified by noting that the boundary terms in $C_1$ and $C_2$ should be negligible at large $N$ and $P$, while $C_3$ and $C_4$ should give a Gaussian contribution at sufficiently large $P$. We now observe that, for any $f_S$ and $f_A$, the conditional means of each term are

$$\mathbb{E}[C_1\,|\,\xi_1^1] = \mathbb{E}[f_S(\Xi)] \tag{F.14}$$

$$\mathbb{E}[C_2\,|\,\xi_1^1] = \lambda\xi_1^1\mathbb{E}[f_A(\Xi)] \tag{F.15}$$

$$\mathbb{E}[C_3\,|\,\xi_1^1] = 0 \tag{F.16}$$

$$\mathbb{E}[C_4\,|\,\xi_1^1] = 0, \tag{F.17}$$

where we note that all $\Xi^\mu$s are identically distributed, so we can simply write $\Xi$ for any one of them. Then, the conditional mean of the crosstalk is

$$\mathbb{E}[C\,|\,\xi_1^1] = \xi_1^1 + \sum_{j=1}^{4}\mathbb{E}[C_j\,|\,\xi_1^1] \tag{F.18}$$

$$= \xi_1^1\{1 + \lambda\mathbb{E}[f_A(\Xi)]\} + \mathbb{E}[f_S(\Xi)]. \tag{F.19}$$

Considering the variance of $C$, the variances of the different contributions are

$$\text{var}[C_1\,|\,\xi_1^1] = \text{var}[f_S(\Xi)] + \lambda^2\mathbb{E}[f_A(\Xi)^2] \tag{F.20}$$

$$\text{var}[C_2\,|\,\xi_1^1] = \mathbb{E}[f_S(\Xi)^2] + \lambda^2\,\text{var}[f_A(\Xi)] \tag{F.21}$$

$$\text{var}[C_3\,|\,\xi_1^1] = (P-3)\mathbb{E}[f_S(\Xi)^2] \tag{F.22}$$

$$\text{var}[C_4\,|\,\xi_1^1] = \lambda^2(P-3)\mathbb{E}[f_A(\Xi)^2], \tag{F.23}$$

while the covariances are

$$\text{cov}[C_1, C_2\,|\,\xi_1^1] = 0 \tag{F.24}$$

$$\text{cov}[C_1, C_3\,|\,\xi_1^1] = \lambda\mathbb{E}[f_A(\Xi)]\mathbb{E}[f_S(\Xi)] \tag{F.25}$$

$$\text{cov}[C_1, C_4\,|\,\xi_1^1] = 0, \tag{F.26}$$

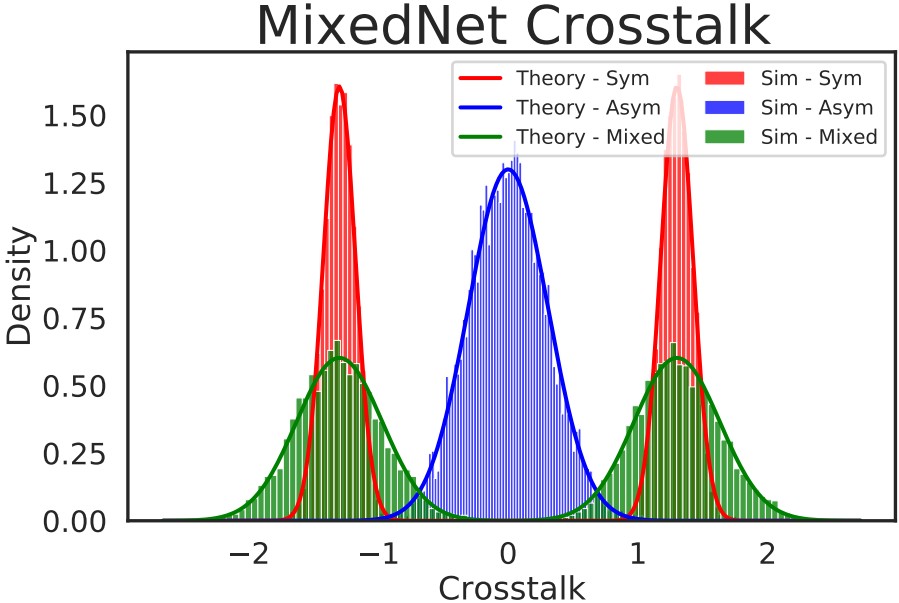

Figure F.1: Crosstalk of Polynomial `MixedNet` where $N = 100$, $\lambda = 2.5$, $d_S = d_A = 3$ and $P = 1000$ patterns are stored. Histograms are generated for patterns drawn from $5000$ randomly sequences and theoretical curves are plotted. Green represents the full crosstalk for the `MixedNet`. Blue and red represent the asymmetric and symmetric terms of the crosstalk, respectively. Observe that the bimodality in the full model comes from bimodality in the symmetric term.

$$\text{cov}[C_2, C_3 \,|\, \xi_1^1] = 0 \tag{F.27}$$

$$\text{cov}[C_2, C_4 \,|\, \xi_1^1] = \lambda \mathbb{E}[f_S(\Xi)] \mathbb{E}[f_A(\Xi)], \tag{F.28}$$

and

$$\text{cov}[C_3, C_4 \,|\, \xi_1^1] = \lambda \sum_{\mu,\nu=3}^{P-1} \mathbb{E}[\xi_1^\mu \Xi_1^{\nu+1}] \mathbb{E}[f_S(\Xi^\mu) f_A(\Xi^\nu)] \tag{F.29}$$

$$= \lambda \sum_{\mu=3}^{P} \mathbb{E}[f_S(\Xi^\mu)] \mathbb{E}[f_A(\Xi^{\mu-1})] \tag{F.30}$$

$$= \lambda(P-3) \mathbb{E}[f_S(\Xi)] \mathbb{E}[f_A(\Xi)]. \tag{F.31}$$

Therefore, the conditional variance of the crosstalk is

$$\text{var}[C \,|\, \xi_1^1] = \sum_{j=1}^{4} \text{var}[C_j \,|\, \xi_1^1] + 2 \sum_{j=1}^{4} \sum_{k>j} \text{cov}[C_j, C_k \,|\, \xi_1] \tag{F.32}$$

$$= (P-3)\{\mathbb{E}[f_S(\Xi)^2] + 2\lambda \mathbb{E}[f_S(\Xi)]\mathbb{E}[f_A(\Xi)] + \lambda^2 \mathbb{E}[f_A(\Xi)^2]\}$$
$$+ \text{var}[f_S(\Xi)] + \lambda^2 \mathbb{E}[f_A(\Xi)^2] + \mathbb{E}[f_S(\Xi)^2] + \lambda^2 \text{var}[f_A(\Xi)] + 4\lambda \mathbb{E}[f_A(\Xi)]\mathbb{E}[f_S(\Xi)] \tag{F.33}$$

$$= (P-1)\{\mathbb{E}[f_S(\Xi)^2] + 2\lambda \mathbb{E}[f_S(\Xi)]\mathbb{E}[f_A(\Xi)] + \lambda^2 \mathbb{E}[f_A(\Xi)^2]\}$$
$$- \mathbb{E}[f_S(\Xi)]^2 - \lambda^2 \mathbb{E}[f_A(\Xi)]^2. \tag{F.34}$$

For large $P$ and $N$, the two terms on the second line of this result will be subleading, as they do not scale with $P$ and have identical or subleading scaling with $N$ to the terms that do scale with $P$. That is, we have

$$\text{var}[C \,|\, \xi_1^1] \sim P\{\mathbb{E}[f_S(\Xi)^2] + 2\lambda \mathbb{E}[f_S(\Xi)]\mathbb{E}[f_A(\Xi)] + \lambda^2 \mathbb{E}[f_A(\Xi)^2]\}. \tag{F.35}$$

Collecting these results, we have

$$\mathbb{E}[C \mid \xi_1^1] = \xi_1^1 \{1 + \lambda \mathbb{E}[f_A(\Xi)]\} + \mathbb{E}[f_S(\Xi)] \tag{F.36}$$

and

$$\mathrm{var}[C \mid \xi_1^1] \sim P\{\mathbb{E}[f_S(\Xi)^2] + 2\lambda \mathbb{E}[f_S(\Xi)]\mathbb{E}[f_A(\Xi)] + \lambda^2 \mathbb{E}[f_A(\Xi)^2]\}. \tag{F.37}$$

By the law of total probability, we have

$$\mathbb{P}[T_{MN}(\boldsymbol{\xi}^1)_1 \neq \xi_1^2] = \mathbb{P}[C < -\lambda] \tag{F.38}$$

$$= \frac{1}{2}\mathbb{P}[C < -\lambda \mid \xi_1^1 = -1] + \frac{1}{2}\mathbb{P}[C < -\lambda \mid \xi_1^1 = +1] \tag{F.39}$$

$$\sim \frac{1}{2}H\left(\frac{\lambda + \mathbb{E}[C \mid \xi_1^1 = -1]}{\sqrt{\mathrm{var}[C \mid \xi_1^1 = -1]}}\right) + \frac{1}{2}H\left(\frac{\lambda + \mathbb{E}[C \mid \xi_1^1 = +1]}{\sqrt{\mathrm{var}[C \mid \xi_1^1 = +1]}}\right) \tag{F.40}$$

under the bimodal Gaussian approximation to the crosstalk distribution. To have $\mathbb{P}[T_{MN}(\boldsymbol{\xi}^1)_1 \neq \xi_1^2]$, both of these conditional probabilities must tend to zero. By basic concentration arguments, we expect to have

$$\mathbb{E}[C \mid \xi_1^1] \sim \xi_1^1 \tag{F.41}$$

up to corrections that are small in an absolute sense. Moreover, we have

$$\mathbb{E}[C \mid \xi_1^1 = +1] - \mathbb{E}[C \mid \xi_1^1 = -1] = 2\{1 + \lambda \mathbb{E}[f_A(\Xi)]\} \tag{F.42}$$

which for the separation functions considered here is strictly positive. As we keep $\lambda$ constant with $N$ and $P$, we must have

$$\mathbb{E}[C \mid \xi_1^1 = -1] > -\lambda \tag{F.43}$$

and $\mathrm{var}[C \mid \xi_1^1 = -1] \to 0$ in order to have $\mathbb{P}[T_{MN}(\boldsymbol{\xi}^1)_1 \neq \xi_1^2] \to 0$. But, given the formula above, $\mathrm{var}[C \mid \xi_1^1 = -1] = \mathrm{var}[C \mid \xi_1^1 = +1]$, so this implies that the $\xi_1^1 = +1$ contribution to the probability will be exponentily suppressed. hen, we can apply an identical argument to that which we used for the DenseNet in Appendix C to obtain the asymptotic behavior of $\mathbb{P}[C < -\lambda \mid \xi_1^1 = -1]$, yielding

$$\mathbb{P}[T_{MN}(\boldsymbol{\xi}^1)_1 \neq \xi_1^2] \sim \frac{1}{2}H\left(\frac{\lambda + \mathbb{E}[C \mid \xi_1^1 = -1]}{\sqrt{\mathrm{var}[C \mid \xi_1^1 = -1]}}\right) \tag{F.44}$$

$$\sim \frac{1}{2\sqrt{2\pi}}\frac{\sqrt{\mathrm{var}[C \mid \xi_1^1 = -1]}}{\lambda + \mathbb{E}[C \mid \xi_1^1 = -1]}\exp\left(-\frac{1}{2}\frac{(\lambda + \mathbb{E}[C \mid \xi_1^1 = -1])^2}{\mathrm{var}[C \mid \xi_1^1 = -1]}\right). \tag{F.45}$$

For this to work, we must clearly have $\lambda > 1$.

We could in principle compute the excess kurtosis of the crosstalk for the MixedNet as we did for the DenseNet, but we will not do so here as the computation would be tedious and would not yield substantial new insight beyond that for the DenseNet.

## F.1 Polynomial MixedNet

We first consider the polynomial mixed network, with $f_S(x) = x^{d_S}$ and $f_A(x) = x^{d_A}$ for two possibly differing degrees $d_S, d_A \in \mathbb{N}_{>0}$. We can apply the same reasoning as in Appendix C.1 to obtain the required moments at large $N$, which yields the first moments

$$\mathbb{E}[f_S(\Xi)] = \mathbb{E}[\Xi^{d_S}] = \begin{cases} 0 & d_S \text{ odd,} \\ \dfrac{(d_S - 1)!!}{N^{d_S/2}}\left[1 + \mathcal{O}\left(\dfrac{1}{N}\right)\right] & d_S \text{ even} \end{cases} \tag{F.46}$$

and

$$\mathbb{E}[f_A(\Xi)] = \mathbb{E}[\Xi^{d_A}] = \begin{cases} 0 & d_A \text{ odd,} \\ \dfrac{(d_A - 1)!!}{N^{d_A/2}}\left[1 + \mathcal{O}\left(\dfrac{1}{N}\right)\right] & d_A \text{ even,} \end{cases} \tag{F.47}$$

and the second moments

$$\mathbb{E}[f_S(\Xi)^2] = \mathbb{E}[\Xi^{2d_S}] = \frac{(2d_S - 1)!!}{N^{d_S}}\left[1 + \mathcal{O}\left(\frac{1}{N}\right)\right] \tag{F.48}$$

and

$$\mathbb{E}[f_A(\Xi)^2] = \mathbb{E}[\Xi^{2d_A}] = \frac{(2d_A - 1)!!}{N^{d_A}}\left[1 + \mathcal{O}\left(\frac{1}{N}\right)\right]. \tag{F.49}$$

Then, the conditional mean of the crosstalk is given by

$$\mathbb{E}[C \,|\, \xi_1^1] \sim \xi_1^1 \tag{F.50}$$

up to corrections which vanish in an absolute, not a relative, sense, while the conditional variance is asymptotic to

$$\mathrm{var}[C \,|\, \xi_1^1] \sim P\left\{\frac{(2d_S - 1)!!}{N^{d_S}} + 2\lambda\frac{(d_S - 1)!!\,(d_A - 1)!!}{N^{(d_S + d_A)/2}}\mathbf{1}\{d_S, d_A \text{ even}\} + \lambda^2\frac{(2d_A - 1)!!}{N^{d_A}}\right\}. \tag{F.51}$$

We must now determine the storage capacity. We recall that, in all case, we want $P$ to tend to infinity slowly enough that $\mathrm{var}[C \,|\, \xi_1^1]$ tends to zero. Then, we can see that what matters is which of the terms inside the curly brackets in the expression for the conditional variance above tends to zero with $N$ the slowest. This is of course determined by $\min\{d_S, d_A\}$, but the constant factor multiplying the leading term will depend on which is smaller, or if they are equal. First, consider the case in which $d_S = d_A = d$. Then, we have

$$\mathrm{var}[C \,|\, \xi_1^1] \sim \frac{P}{N^d}\left\{(2d - 1)!! + 2\lambda(d - 1)!!\,(d - 1)!!\mathbf{1}\{d \text{ even}\} + \lambda^2(2d - 1)!!\right\}. \tag{F.52}$$

Now, consider the case in which $d_S < d_A$. Then, $(d_S + d_A)/2 > d_S$, hence the $N^{-d_S}$ term dominates and we have

$$\mathrm{var}[C \,|\, \xi_1^1] \sim \frac{P}{N^{d_S}}(2d_S - 1)!!. \tag{F.53}$$

Similarly, if $d_A > d_S$, the $N^{-d_A}$ term dominates, and we have

$$\mathrm{var}[C \,|\, \xi_1^1] \sim \frac{P}{N^{d_A}}\lambda^2(2d_A - 1)!!. \tag{F.54}$$

We can summarize these results as

$$\mathrm{var}[C \,|\, \xi_1^1] \sim \gamma_{d_S, d_A}\frac{P}{N^{\min\{d_S, d_A\}}}, \tag{F.55}$$

where

$$\gamma_{d_S, d_A} = \begin{cases} (2d_S - 1)!! & \text{if } d_S < d_A, \\ (\lambda^2 + 1)(2d_S - 1)!! + 2\lambda[(d_S - 1)!!]^2\mathbf{1}\{d_S \text{ even}\} & \text{if } d_S = d_A, \\ \lambda^2(2d_A - 1)!! & \text{if } d_S > d_A. \end{cases} \tag{F.56}$$

Using the general arguments presented above, we then have

$$\mathbb{P}[T_{MN}(\boldsymbol{\xi}^1)_1 \neq \xi_1^2] \sim \frac{1}{2\sqrt{2\pi}}\sqrt{\frac{\gamma_{d_S, d_A}P}{(\lambda - 1)^2 N^{\min\{d_S, d_A\}}}}\exp\left(-\frac{(\lambda - 1)^2}{2}\frac{N^{\min\{d_S, d_A\}}}{\gamma_{d_S, d_A}P}\right). \tag{F.57}$$

for any $\lambda > 1$. We must first determine the single-transition capacity, which requires that $N\mathbb{P}[T_{MN}(\boldsymbol{\xi}^1)_1 \neq \xi_1^2] \to 0$. Recalling that our argument requires us to take $P \to \infty$ slowly enough that $\mathrm{var}[C \,|\, \xi_1^1] \to 0$, we make the *Ansatz* that

$$P \sim \frac{(\lambda - 1)^2}{\alpha\gamma_{d_S, d_A}}\frac{N^{\min\{d_S, d_A\}}}{\log N} \tag{F.58}$$

for some $\alpha$. This yields

$$N\mathbb{P}[T_{MN}(\boldsymbol{\xi}^1)_1 \neq \xi_1^2] \sim \frac{1}{2\sqrt{2\pi\alpha\log N}} N^{1-\alpha/2}, \tag{F.59}$$

which tends to zero if $\alpha \geq 2$, yielding a predicted capacity of

$$P_T \sim \frac{(\lambda-1)^2}{2\gamma_{d_S,d_A}} \frac{N^{\min\{d_S,d_A\}}}{\log N}. \tag{F.60}$$

We now consider the sequence capacity, which requires that $NP\mathbb{P}[T_{MN}(\boldsymbol{\xi}^1)_1 \neq \xi_1^2] \to 0$. Then, making the same *Ansatz* for $P$ as above, we have

$$NP\mathbb{P}[T_{MN}(\boldsymbol{\xi}^1)_1 \neq \xi_1^2] \sim \frac{1}{2\sqrt{2\pi}} \frac{(\lambda-1)^2}{\gamma_{d_S,d_A}} \frac{1}{(\alpha\log N)^{3/2}} N^{\min\{d_S,d_A\}+1-\alpha/2}, \tag{F.61}$$

which tends to zero provided that $\alpha \geq 2(\min\{d_S, d_A\} + 1)$, yielding a predicted capacity of

$$P_S \sim \frac{(\lambda-1)^2}{2(\min\{d_S,d_A\}+1)\gamma_{d_S,d_A}} \frac{N^{\min\{d_S,d_A\}}}{\log N}. \tag{F.62}$$

## F.2 Exponential `MixedNet`

We now consider the Exponential `MixedNet`, with $f_S(x) = f_A(x) = e^{(N-1)(x-1)}$. With this, we have the first moments

$$\mathbb{E}[f_S(\Xi)] = \mathbb{E}[f_A(\Xi)] = \exp[-(N-1)]\mathbb{E}\left[\exp\left(\sum_{j=2}^{N} \xi_j\right)\right] \tag{F.63}$$

$$= \exp[-(N-1)]\prod_{j=2}^{N} \mathbb{E}[\exp(\xi_j)] \tag{F.64}$$

$$= \left(\frac{\cosh(1)}{\exp(1)}\right)^{N-1} \tag{F.65}$$

and the second moments

$$\mathbb{E}[f_S(\Xi)^2] = \mathbb{E}[f_A(\Xi)^2] = \left(\frac{\cosh(2)}{\exp(2)}\right)^{N-1} = \frac{1}{\beta^{N-1}}, \tag{F.66}$$

where as in Appendix C.2 we let

$$\beta = \frac{\exp(2)}{\cosh(2)} \simeq 1.96403. \tag{F.67}$$

Noting that

$$\frac{\exp(1)}{\cosh(1)} \simeq 1.76159, \tag{F.68}$$

the conditional mean of the crosstalk is then

$$\mathbb{E}[C\,|\,\xi_1^1] = \xi_1^1\{1 + \lambda\mathbb{E}[f_A(\Xi)]\} + \mathbb{E}[f_S(\Xi)] \tag{F.69}$$

$$= \xi_1^1\left\{1 + \lambda\left(\frac{\cosh(1)}{\exp(1)}\right)^{N-1}\right\} + \left(\frac{\cosh(1)}{\exp(1)}\right)^{N-1} \tag{F.70}$$

$$\sim \xi_1^1, \tag{F.71}$$

where the corrections are exponentially small in an absolute sense. The leading part of the conditional variance of the crosstalk is

$$\text{var}[C\,|\,\xi_1^1] \sim P\{\mathbb{E}[f_S(\Xi)^2] + 2\lambda\mathbb{E}[f_S(\Xi)]\mathbb{E}[f_A(\Xi)] + \lambda^2\mathbb{E}[f_A(\Xi)^2]\} \tag{F.72}$$

$$\sim \frac{P}{\beta^{N-1}}\left\{1 + 2\lambda\left(\frac{\cosh(1)^2}{\cosh(2)}\right)^{N-1} + \lambda^2\right\} \tag{F.73}$$

$$\sim \frac{P}{\beta^{N-1}}(1 + \lambda^2), \tag{F.74}$$

S23

where we note that

$$\frac{\cosh(1)^2}{\cosh(2)} \simeq 0.632901 \tag{F.75}$$

hence the other contribution is exponentially suppressed in a relative sense.

We thus have

$$\mathbb{E}[C \,|\, \xi_1^1] \sim \xi_1^1 \tag{F.76}$$

$$\mathrm{var}[C \,|\, \xi_1^1] \sim \frac{P}{\beta^{N-1}}(1 + \lambda^2), \tag{F.77}$$

hence from the general argument above we have

$$\mathbb{P}[T_{MN}(\boldsymbol{\xi}^1)_1 \neq \xi_1^2] \sim \frac{1}{2\sqrt{2\pi}}\frac{\sqrt{(1+\lambda^2)}}{\lambda-1}\sqrt{\frac{P}{\beta^{N-1}}}\exp\left(-\frac{1}{2}\frac{(\lambda-1)^2}{1+\lambda^2}\frac{\beta^{N-1}}{P}\right) \tag{F.78}$$

for $\lambda > 1$. We now want to determine the capacity, starting with the single-transition capacity, for which we must have $N\mathbb{P}[T_{MN}(\boldsymbol{\xi}^1)_1 \neq \xi_1^2] \to 0$. Recalling that we want to have $\mathrm{var}[C \,|\, \xi_1^1] \to 0$, we make the *Ansatz*

$$P \sim \frac{1}{\alpha}\frac{(\lambda-1)^2}{\lambda^2+1}\frac{\beta^{N-1}}{\log N} \tag{F.79}$$

for some $\alpha$, which yields

$$N\mathbb{P}[T_{MN}(\boldsymbol{\xi}^1)_1 \neq \xi_1^2] \sim \frac{1}{2\sqrt{2\pi\alpha\log N}}N^{1-\alpha/2}. \tag{F.80}$$

This tends to zero if $\alpha \geq 2$, hence we conclude that the Gaussian theory predicts

$$P_T \sim \frac{1}{2}\frac{(\lambda-1)^2}{\lambda^2+1}\frac{\beta^{N-1}}{\log N}. \tag{F.81}$$

We now want to determine the sequence capacity, which requires that $NP\mathbb{P}[T_{MN}(\boldsymbol{\xi}^1)_1 \neq \xi_1^2] \to 0$. Following our analysis of the Exponential `DenseNet` in Appendix C.2, we make the *Ansatz* that

$$P \sim \frac{1}{\alpha}\frac{(\lambda-1)^2}{\lambda^2+1}\frac{\beta^{N-1}}{N}, \tag{F.82}$$

which yields

$$NP\mathbb{P}[T_{MN}(\boldsymbol{\xi}^1)_1 \neq \xi_1^2] \sim \frac{1}{2\sqrt{2\pi\alpha N}}\frac{1}{\alpha\beta}\frac{(\lambda-1)^2}{\lambda^2+1}\exp\left[\left(\log\beta - \frac{\alpha}{2}\right)N\right]. \tag{F.83}$$

This tends to zero if $\alpha \geq 2\log\beta$, giving a predicted sequence capacity of

$$P_S = \frac{1}{2\log\beta}\frac{(\lambda-1)^2}{\lambda^2+1}\frac{\beta^{N-1}}{N}. \tag{F.84}$$

Thus, for both definitions of capacity, the Gaussian theory's prediction of the capacity of the Exponential `MixedNet` is

$$\frac{(\lambda-1)^2}{\lambda^2+1} \tag{F.85}$$

times the capacity of the Exponential `DenseNet` analyzed in Appendix C.2. This factor tends to zero from above as $\lambda \downarrow 1$, and gradually increases to 1 as $\lambda \to \infty$. Note that even without explicitly computing the excess kurtosis, we expect the intuition from the Exponential `DenseNet` to carry over to this setting. Indeed, the numerical simulations in Figure F.2 show that the transition capacity is well captured by the Gaussian theory while the sequence capacity shows significant deviation for small `MixedNets`.

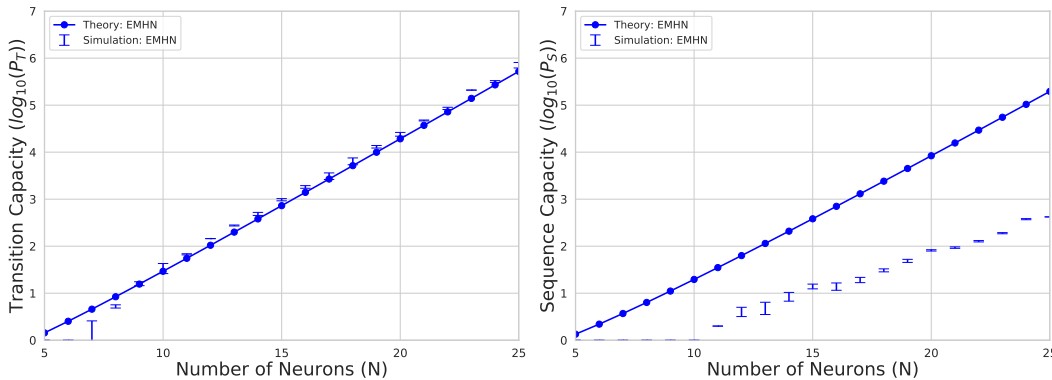

Figure F.2: The capacities of Exponential `MixedNets` with $\lambda = 2.5$ are plotted as a function of network size. (A) Transition capacity for the Exponential `MixedNet`, which closely matches theoretical prediction. The predicted capacity is shown by the solid line with dots, while square error bars show the results of numerical experiment. (B) Sequence capacity for the Exponential `MixedNet`, which diverges from theoretical prediction.

## G   Numerical implementation

Source code is available on GitHub at https://github.com/Pehlevan-Group/LongSequenceHopfieldMemory. Experiments were run on the Harvard University FAS RC Cannon HPC cluster (https://www.rc.fas.harvard.edu/), using Nvidia A100 80GB GPUs. This limited the maximum number of patterns we could store in memory simultaneously to approximately $10^6$ patterns, restricting our experimental evaluation of the Exponential `DenseNet` to approximately $N = 25$ neurons.

### G.1   Transition capacity

Numerical simulations for transition capacity were conducted as follows: For a given model of size $N$, start by initializing 100 sequences of Rademacher distributed patterns of length $P_0$, where $P_0 = 2P^*$ is well above the model's predicted capacity $P^*$. This initialization for $P_0$ was found through trial and error, where the method detects if you start below capacity. The model's update rule is applied in parallel across all patterns and across all sequences. If errors are made for any pattern in any sequence, 100 new random sequences are generated with smaller length $P_1 = 0.99P_0$. This is repeated, with the new sequence length being $P_{t+1} = 0.99P_t$, until 100 sequences are generated for which no error is made in any transition. This entire process is repeated 20 times starting from $P_0$ in order to obtain error bars.

### G.2   Sequence capacity

Numerical simulations for sequence capacity were conducted in a similar fashion. For a given model of size $N$, start by initializing 100 sequences of Rademacher distributed patterns of length $P_0$, where $P_0$ is well above the model's capacity. Starting from the first pattern of each sequence, the model's update rule is applied serially for each sequence. As soon as an error is obtained within any sequence, 100 new random sequences are generated with smaller length $P_1 = 0.99P_0$. This is repeated, with the new sequence length being $P_{t+1} = 0.99P_t$, until 100 sequences are generated for which no error is made. This entire process is repeated 20 times starting from $P_0$ in order to obtain error bars.

### G.3   MovingMNIST

For the MovingMNIST experiments in Section 2.4, the images were pre-processed to have binarized pixel values. There were 10000 subsequences, each containing 2 handwritten digits from the MNIST dataset moving through each other across 20 images, that were concatenated to construct the entire sequence of 200000 images [78]. Then, different models were run from initialization and their output for different time steps was displayed in Figure 3.

### G.4   Generalized pseudoinverse rule

For numerical simulations of the generalized pseudoinverse rule in 2.5, the transition capacity of the Polynomial `DenseNet` was simulated in a similar method as described above. However, the Exponential `DenseNet` suffered from numerical instability when calculating the pseudoinverse of the overlap matrix, resulting in floating point error. Therefore, we showed results only for the Polynomial `DenseNet`.

