# OpenReview forum: "Long Sequence Hopfield Memory"
_NeurIPS.cc/2023/Conference — NeurIPS 2023 poster_

### Official Review · Reviewer_2PbW · 2023-06-16

**Soundness:** 3 good
**Presentation:** 3 good
**Contribution:** 3 good
**Rating:** 6
**Confidence:** 4

**Summary:**

This paper proposes to introduce nonlinear interaction terms in the Amari-Hopfield type recurrent networks for learning long binary sequences. Analytical results on the network capacity and a new learning algorithm are provided.

**Strengths:**

1. The analytic calculation of the network capacity on random sequences in Section 2 is a very interesting theoretical result, extending previous work on the capacity of storing static patterns in the networks. The simulation results match the theoretical calculation, which is excellent.
2. The paper is well-written and easy to follow.
3. Code is provided for the reproduction of the experiments.

**Weaknesses:**

1. It is technically incorrect to state that "a major limitation of the Hopfield Network and related associative memory models is its capacity". In [1], it has been shown Hopfield model can store sequences of maximal length 2^N. The short length of sequences is largely due to the temporal asymmetric Hebbian learning algorithm, rather than the model.

2. There is a vast of papers [2,3,4,5] on overcoming the limitation of storing correlated in Hopfield model by the Hebbian algorithm, which are not mentioned this paper. The ideas can be extended to store sequences. The authors should do a literature review on them and compare the pros and cons with their approach in solving the problem.

3. The theory is conducted on random sequences, rather than the correlated pattern sequences which should be addressed. In the experiments in Section 2.4, the MNIST digits images in a sequence are in essentially random orders. Therefore the correlation of patterns is not strong enough to validate the propose method.

4. Robustness of sequence retrieval is missing in the paper. A major feature of Hopfield model is its robustness of recovering static patterns under noise. Can the proposed method recover the rest of a sequence, given a perturbed pattern? An evaluation is needed.

5. In Line 65-68, the $\xi$ should be bold-faced. In Line 95, $\xi_2^1$ should be $\xi_1^2$.

[1] Exponentially Long Orbits in Hopfield Neural Networks, Muscinelli, Gerstner, Brea. Neural Computation, 2017.
[2] Learning of correlated patterns in spin-glass networks by local learning rules. Diederich, Opper. PRL, 1987.
[3] The space of interactions in neural network models. Gardner. Journal of Physics A, 1988.
[4] Perceptron-like learning in time-summating neural networks. Bressloff, Taylor. Journal of Physics A, 1999.
[5] Dynamics of Memory Representations in Networks with Novelty-Facilitated Synaptic Plasticity. Blumenfeld, Preminger, Sagi, Tsodyks. Neuron, 2006.

**Questions:**

Overall, I appreciate the theoretical part of this paper, which is valuable to the community per se. However, I found it is not consistent with the problem addressed in the beginning of this paper, learning long and correlated pattern sequences. The experiments do not convince me that correlated pattern sequences can be handled by the proposed method either.

However, I would like to raise my rating by 1-2 points, if the authors can in the rebuttal or revised paper
1) evaluate their method on the Moving MNIST dataset, in which the successive patterns are highly correlated.
https://www.cs.toronto.edu/~nitish/unsupervised_video/

2) evaluate their method for the robustness of sequence retrieval under noise.

Moreover, if the authors can additionally provide some theoretical justification on storing correlated sequence patterns (a shortcoming which I have pointed out in the review), plus all my concerns are well-addressed, I could raise the rating by 3-4 points.


### Rebuttal
I have read the rebuttal and believe the authors have addressed my points. I therefore raise my rating to "6. Weakly Accept".



**Limitations:**

Yes, the limitation of the analytic calculation is mentioned in Line 353-358.

---

> ### Author Rebuttal · Authors · 2023-08-09
>
> Thank you for the clear criticism and insightful comments, especially the suggestion to use the Moving MNIST dataset and include simulations demonstrating robust sequence retrieval, which we have included the global rebuttal. We have also sketched an outline for the capacity of biased patterns.
>
> **Weaknesses:**
> > It is technically incorrect to state that "a major limitation of the Hopfield Network and related associative memory models is its capacity". In [1], it has been shown Hopfield model can store sequences of maximal length 2^N. The short length of sequences is largely due to the temporal asymmetric Hebbian learning algorithm, rather than the model.
>
> Thank you for this comment. By Hopfield Network, we specifically refer to a network with a Hebbian-like learning rule, and will clarify this in the camera-ready version. The paper mentioned in [1] describes an algorithm such that given a network of size $N$, one can construct a sequence of $2^N$ patterns and a specific update rule to recall these patterns. However, it will not work for arbitrary sequences and does not have error correction capabilities, which is a core feature of our model. We will be more precise in the camera-ready version.
>
> > There is a vast of papers [2,3,4,5] on overcoming the limitation of storing correlated in Hopfield model by the Hebbian algorithm, which are not mentioned this paper. The ideas can be extended to store sequences. The authors should do a literature review on them and compare the pros and cons with their approach in solving the problem.
>
> This is a fair point. We tried to do a brief survey in the introduction of the paper, but the literature on associative memory is vast and some papers were accidentally omitted. We have included a brief comparison to other methods in the global rebuttal and will include a deeper one in the camera-ready version, alongside citing these papers. If you believe any other models are relevant, we would be grateful.
>
> > The theory is conducted on random sequences, rather than the correlated pattern sequences which should be addressed. In the experiments in Section 2.4, the MNIST digits images in a sequence are in essentially random orders. Therefore the correlation of patterns is not strong enough to validate the propose method.
>
> Thank you for this suggestion. We ran simulations on the Moving MNIST dataset which has higher overlap between patterns and included the results in the global rebuttal.
>
> > Robustness of sequence retrieval is missing in the paper. A major feature of Hopfield model is its robustness of recovering static patterns under noise. Can the proposed method recover the rest of a sequence, given a perturbed pattern? An evaluation is needed.
>
> This is great suggestion, and we have ran simulations which we've further discussed in the global rebuttal. In Figure 3, we add noise to each transition in the sequence which would be even more difficult than only perturbing the first pattern.
>
> > In Line 65-68, the $\xi$ should be bold-faced. In Line 95, $\xi_2^1$ should be $\xi_1^2$.
>
> Thank you for pointing that out; we have fixed these typos.
>
> **Questions:**
> > Overall, I appreciate the theoretical part of this paper, which is valuable to the community per se. However, I found it is not consistent with the problem addressed in the beginning of this paper, learning long and correlated pattern sequences. The experiments do not convince me that correlated pattern sequences can be handled by the proposed method either. However, I would like to raise my rating by 1-2 points, if the authors can in the rebuttal or revised paper:
>     1. evaluate their method on the Moving MNIST dataset, in which the successive patterns are highly correlated. https://www.cs.toronto.edu/~nitish/unsupervised_video/
>     2. evaluate their method for the robustness of sequence retrieval under noise.
>
> Thank you for the valuable suggestions, and we appreciate you pointing us to the Moving MNIST dataset of which we were not previously aware. We have ran both of these simulations and included them in the global rebuttal.
>
> > Moreover, if the authors can additionally provide some theoretical justification on storing correlated sequence patterns (a shortcoming which I have pointed out in the review), plus all my concerns are well-addressed, I could raise the rating by 3-4 points.
>
> Thank you for this question. We agree that it would be very interesting to extend our theoretical arguments to correlated patterns. Our approximate Gaussian computations should be easily extensible to the case in which the patterns are biased (not correlated in the statistical sense, but leading to non-zero average overlap, which is the relevant notion in this context); the effect would be to shift the mean and variance of the crosstalk distributions. Concretely, consider patterns with
> $$\mathbb{P}[\xi_{j}^{\mu} = \pm 1] = \frac{1 \pm r}{2},$$
>  such that
> $$\mathbb{E}[\xi_{j}^{\mu}] = r$$
> is the bias. This setup mirrors that considered in the classic work of Gardner (JPA 1988), which has been extended to simple nonlinear networks (Zavatone-Veth & Pehlevan PRE 2021); for that unconstrained learning rule the bias does not alter the scaling of capacity with $N$. We propose to do this computation for the final version of our paper, as it should not be challenging.
>
> To sketch the computation, we recall that our goal is to analyze the crosstalk and approximate its distribution as Gaussian.
> $$ C = \sum_{\mu=2}^P \xi^{2}_{1} \xi_1^{\mu+1} f \left( X^{\mu} \right), $$
>
> where
> $$ X^{\mu} = \frac{1}{N-1} \sum_{j=2}^{N} \xi_{j}^{\mu} \xi_{j}^{1}$$
> Since the patterns are no longer symmetrically distributed, we must now also track the non-zero mean of the crosstalk. However, the computation is still reasonably straightforward.

---

> > ### Comment · Reviewer_2PbW · 2023-08-11
> >
> > I have read the rebuttal and believe the authors have addressed my points. I therefore raise my rating to "6. Weakly Accept".
> >
> > However, the authors should provide more quantitative empirical analysis for the capacity/robustness trade-off. I hope the authors can add this in the final version of the paper.

---

> > > ### Author Response · Authors · 2023-08-13
> > >
> > > Thank you for your suggestions and the updated score. We will certainly include some more empirical analysis for the capacity/robustness trade-off in the final version. We have already ran simulations where we examine the effect of a non-zero error threshold when computing the transition and sequence capacities.

---

### Official Review · Reviewer_6yTi · 2023-07-05

**Soundness:** 2 fair
**Presentation:** 4 excellent
**Contribution:** 3 good
**Rating:** 6
**Confidence:** 4

**Summary:**

This paper combines two modifications to the basic Hopfield network to create a network capable of creating and recalling long sequential memories (i.e. sequences of states of the network). The sequence capacity of the proposed network is bounded analytically, and supported by numerical simulations. This model is then extended to allow memories to be recalled a fixed number of times before transitioning to the next item in the sequence. Finally, an implementation of this network using more biologically plausible neurons is given.

**Strengths:**

* This work extends the basic Hopfield network in several important directions while retaining the original model's simplicity and (to some extent) analytic tractability.
* The paper itself is well-written and eminently readable, and is likely to have broad appeal in related disciplines. Although it covers several ideas related to sequential Hopfield networks, the paper still feels cohesive and flows well.
* The theory and simulations complement each other well.

**Weaknesses:**

* The theoretical analysis crucially relies on nonrigorously approximating a nonlinear function of i.i.d. random variables as Gaussian. The authors make a substantial attempt to provide justification for this, but this does dilute the utility of the theoretical analysis.

**Questions:**

* Can you elaborate on the relationship between the models presented here and the Karuvally et al. work mentioned on line 46?
* What kind of loss in sharpness results if you circumvent the Gaussian approximation (lines 98-105) by using McDiarmid's inequality to bound the tail instead? I believe this should also allow the capacity to be bounded for any $L$-Lipschitz nonlinearity.
* In lines 201-213, it seems like the patterns are not actually "correlated", as each pattern is still i.i.d. but the distribution is no longer uniform. If so, care should be taken to distinguish this meaning from the term "correlation" as it is used in Section 2.4, meaning that the patterns are not i.i.d. On that note, a synthetic version of the MNIST experiment where there is a ground truth sequence, and different sequences are generated by flipping each of its bits with some small probability, would be interesting.

**Limitations:**

Yes

---

> ### Author Rebuttal · Authors · 2023-08-09
>
> Thank you for the comments and suggestions, especially those on improving the rigor of the technical analysis. Since the original submissions, we have proven rigorous bounds for the Polynomial DenseNet and are working on rigorous bounds for the Exponential DenseNet, which has proven to be substantially more difficult.
>
> **Weaknesses:**
> > The theoretical analysis crucially relies on nonrigorously approximating a nonlinear function of i.i.d. random variables as Gaussian. The authors make a substantial attempt to provide justification for this, but this does dilute the utility of the theoretical analysis.
>
> Thank you for this comment; as we acknowledge the analysis presented is approximate. A small clarification: the crosstalk is not simply a nonlinear function of a sum of i.i.d. random variables, but a sum of functions of i.i.d. sums of i.i.d. random variables. For the Polynomial DenseNet, we have in hand a rigorous analysis that adapts the analysis of the symmetric MHN by Demircigil et al. 2017 to bound the bitflip probability. We would be happy to add this to our manuscript if the referee believes it would be helpful. Please see also the discussion below under your question about McDiarmid's inequality.
>
> **Questions:**
> > Can you elaborate on the relationship between the models presented here and the Karuvally et al. work mentioned on line 46?
>
> Karuvally et al. 2022 takes an interesting approach toward extending Modern Hopfield Networks to store sequences. While we forego the energy function formulation altogether, they develop a way to analyze an energy function in terms of an adiabatic limit. We believe that their model, GSEMM, can most closely be described as the biologically-plausible implementation of our proposed Mixed Network. Furthermore, they analyze the model in the setting of continuous-time dynamics, allow for different timescales between the hidden and feature layers, and also allow for intralayer synapses within the hidden layer. On the other hand, our paper is focused on providing a theoretical analysis of the model and its sequence capacity, alongside introducing the generalized pseudoinverse rule to store correlated patterns. Karuvally et al. 2022, however, do not derive an analytic expression for the sequence capacity, which is the central result in our paper.
>
> > What kind of loss in sharpness results if you circumvent the Gaussian approximation (lines 98-105) by using McDiarmid's inequality to bound the tail instead? I believe this should also allow the capacity to be bounded for any Lipschitz nonlinearity.
>
> Thank you for the suggestion. We have in fact previously attempted to estimate the capacity by using McDiarmid's inequality to bound the bitflip probability. For the polynomial DenseNet, the sensitivity to flipping one of the pattern elements inside the nonlinear separation function led to a bound of the form $\mathbb{P}[C>1] \leq [1+o(1)] \exp[-n/(2p)]$, which only recovers the capacity of the classic SeqNet. If the reviewer has any idea for circumventing this loss of sharpness, we would greatly appreciate their feedback. As an alternative approach to rigorously controlling the bitflip probability for the polynomial DenseNet, we believe that the argument used by Demircigil et al. (2017) to prove their theorem on symmetric networks can be adapted step-by-step to prove the Gaussian prediction. We would be happy to include this argument if the reviewer thinks it would enhance our paper. For the exponential DenseNet, the situation is more substantially more difficult due to the tail decaying more slowly,, and we are investigating alternative derivations.
>
> > In lines 201-213, it seems like the patterns are not actually "correlated", as each pattern is still i.i.d. but the distribution is no longer uniform. If so, care should be taken to distinguish this meaning from the term "correlation" as it is used in Section 2.4, meaning that the patterns are not i.i.d.
>
> Thank you for mentioning this, as the imprecise wording here is a historical artifact. Within the Hopfield Network literature, the term "correlated patterns" is often used to refer to "overlapping patterns". As it is the overlap between patterns which increases crosstalk, increasing the overlap will reduce the signal-to-noise ratio when computing sequence transitions. In other words, we are concerned with the raw second moment of the distribution of patterns rather than the centered second moment of the distribution. One can increase the overlap by increasing the bias or by increasing the correlation between the patterns.
>
> > On that note, a synthetic version of the MNIST experiment where there is a ground truth sequence, and different sequences are generated by flipping each of its bits with some small probability, would be interesting.
>
> This is a great suggestion. We assume the intent is to introduce a higher degree of correlation into the network. Due to limited space for figures, we were not able to display this exact experiment but believe that Figures 1 and 2 of the Moving MNIST dataset showcase the model's ability to handle pattern overlap and Figure 3 shows the model's robustness to noise/corruption.

---

> > ### Comment · Reviewer_6yTi · 2023-08-13
> >
> > Thank you for the detailed response and the new experiment! I do think that the rigorous bound would substantially strengthen the paper and I hope that you will include it in future versions.

---

### Official Review · Reviewer_Yq1L · 2023-07-05

**Soundness:** 3 good
**Presentation:** 3 good
**Contribution:** 2 fair
**Rating:** 6
**Confidence:** 4

**Summary:**

This is well presented paper one the storage of sequence memories in Hopfield-like networks. It builds on the recent modern Hopfield networks, but now with asymmetric weights connecting adjacent memories within a sequence. Theoretical capacity limits are calculated and then compared to simulations. Adaptations to improve correlated inputs are proposed using a pseudoinverse rule.

**Strengths:**

Well presented paper. Easy to read. Technically solid.

**Weaknesses:**

1)	This is a small update paper to the recent spate of modern Hopfield network papers (Krotov&Hopfield 2020, Ramsauer et al 2019, etc.,). The proposed method of sequence memory only works when no elements of the sequence are repeated (big limitation), and for unstructured sequences (i.e. it would have no idea if the sequence was drawn from an underlying distributions).

2)	There are also papers that use modern Hopfield networks for sequence memories (e.g. Whittington et al 2022) that don’t have these limitations as they essentially control the sequence retrieval via an external controller. There should be some sort of comparison to existing sequence memory models.

3)	The connection to motor skills is difficult to mesh with what we know about motor learning and episodic memory systems. Episodic memory is a hippocampal phenomenon, whereas motor skills  / learning are the classic example of procedural knowledge.


**Questions:**

Relating to the weaknesses above:
What happens when two patterns are identical in different sequences?
What about when the sequence is drawn from a distribution?


**Limitations:**

I think the main limitations are that this paper is a small advancement from existing modern Hopfield nets., and it’s not clear that the proposed update is a general solution. It is also not clear what the asymmetric weights offers more than other sequence memory models – being that the main point of this paper is that the proposed models is a good model, it really does need some comparisons to existing models. I’m really not saying this proposal is bad - I like it, the presentation is v good, and the theory/simulations concordance is great - it’s just not clear where it stands in the overall landscape, so it’s hard to draw any conclusions.

---

> ### Author Rebuttal · Authors · 2023-08-09
>
> Thank you for your compliments and insightful suggestions. Indeed, this is an extension of existing literature of modern Hopfield networks but we believe that it introduces a general solution for error-correction and the robust retrieval of long sequences. We were primarily focused in this work on theoretically analyzing the model and numerically verifying it via simulation.
>
> **Weaknesses:**
> > This is a small update paper to the recent spate of modern Hopfield network papers (Krotov&Hopfield 2020, Ramsauer et al 2019, etc.,). The proposed method of sequence memory only works when no elements of the sequence are repeated (big limitation), and for unstructured sequences (i.e. it would have no idea if the sequence was drawn from an underlying distributions).
>
> The first part of this statement is correct. Thank you for clarifying it, and we will mention it explicitly in the camera-ready submission. We are actively exploring extensions of the model to store sequences with repeating elements for future work, but we restricted the scope of this paper to non-repeating elements for theoretical tractability. One route we have considered is adding additional associations across longer time steps (e.g. $\xi^{\mu} \to \xi^{\mu+2}$).
>
>
> <!-- However, we do not anticipate this being a major problem in translating the results to represented via contextual embedding which will be  -->
>
> Regarding the second part of the statement, we assume that the sequences are drawn from a Rademacher distribution to compute theoretical capacity as is commonly done in the literature (Amit, Gutfreud, Sompolinsky 1985; McEliece et al. 1987; Krotov, Hopfield 2016). However, the model itself still works when the stored patterns come from a structured underlying distribution, as is demonstrated in the MNIST experiments and Overlapping Patterns experiments in the paper (Figures 5 and 6) and all of the Moving MNIST experiments in the global rebuttal. It simply will have a smaller capacity due to overlap between patterns, although we proposed the generalized pseudoinverse rule as a way to address this.
>
> > There are also papers that use modern Hopfield networks for sequence memories (e.g. Whittington et al 2022) that don’t have these limitations as they essentially control the sequence retrieval via an external controller. There should be some sort of comparison to existing sequence memory models.
>
> We assume the reviewer is referring to "Relating transformers to models and neural representations of the hippocampal formation." This paper was incredibly insightful in connecting the transformer architecture to the Tolman-Eichenbaum machine, and indeed should have been cited when surveying related sequence memory models. However, we believe that this model takes a different approach, in which the positional encoding for each element of the sequence is learned. We assume this is what the reviewer means by "controlling sequence retrieval via an external controller." We were focused on the setting where one would can recall a sequence without access to an external controller. However, while not exactly the same, we briefly suggest an analagous approach for context-dependent gating in the context of motor sequence generation in lines 314-317:
>
> *"In particular, the role of the basal ganglia in this network suggests a novel mechanism of context-dependent gating within Hopfield Networks. Rather than modulating synapses or feature neurons in a network, one can directly inhibit (activate) memory neurons in order to decrease (increase) the likelihood of transitioning to the associated state."*
>
> If there are other connections the reviewer sees, we are open to learning about them and further discussing them.
>
> > The connection to motor skills is difficult to mesh with what we know about motor learning and episodic memory systems. Episodic memory is a hippocampal phenomenon, whereas motor skills / learning are the classic example of procedural knowledge.
>
> Indeed, we do not believe that this model is connected to episodic memory as is found in the hippocampus, but rather are interested in motor sequences that are memorized through repetitive training such as a tennis serve. We seek to point out similarities between the model and recent developments in motor action selection and control, which suggest the role of thalamocortical feedback loops in initiating and automatically preparing motor motifs:
> - Logiaco, Abbott, Escola 2021
> - Kao, Sadabadi, Hennequin 2021
> - Moll et al 2023
>
> We acknowledge that there are differences between our model and the actual neurobiology underlying motor sequence control such as chunking. We will add a sentence to the camera-ready version to clarify this point.
>
> **Limitations:**
> > I think the main limitations are that this paper is a small advancement from existing modern Hopfield nets, and it’s not clear that the proposed update is a general solution. It is also not clear what the asymmetric weights offers more than other sequence memory models – being that the main point of this paper is that the proposed models is a good model, it really does need some comparisons to existing models. I’m really not saying this proposal is bad - I like it, the presentation is v good, and the theory/simulations concordance is great - it’s just not clear where it stands in the overall landscape, so it’s hard to draw any conclusions.
>
> We appreciate the reviewer's comments and concerns. We present a broader survey of related models in the global rebuttal and will modify the introduction in the camera-ready version. If there are other models the reviewer believes should be included in the comparison, we would be happy to investigate them further. Finally, we simply want to reiterate that the focus of the work is the introduction and theoretical analysis of the DenseNet, which we believe provides a general solution for error correction and robust retrieval of long sequences.

---

> > ### Comment · Reviewer_Yq1L · 2023-08-13
> > **Many thanks for the response**
> >
> > Many thanks for the responses. I have appreciated the additional results on moving msnist, as well as appreciating that this is a bigger advance that I had previously thought. I have raised my score accordingly.

---

> > > ### Author Response · Authors · 2023-08-14
> > >
> > > Thank you for your helpful criticism and the updated score.

---

### Official Review · Reviewer_mTmK · 2023-07-06

**Soundness:** 3 good
**Presentation:** 3 good
**Contribution:** 3 good
**Rating:** 6
**Confidence:** 2

**Summary:**

This paper focuses on computational memory that stores sequence data. Existing work that considers Hopfield-like neural networks suffer from limited sequence capacity due to the crosstalk issue. To this end, this paper introduces a nonlinear interaction term inspired by Dense Associative Memories, enhancing pattern separation. The authors develop novel scaling laws for sequence capacity relative to network size, outperforming traditional Hopfield-based models. The authors also propose a generalized pseudoinverse rule for recalling sequences with highly correlated patterns. Additionally, this paper presents a biologically plausible implementation with connections to motor neuroscience to store sequences with variable timing.

**Strengths:**

1. Designing computational memory for sequence data is interesting and novel.

2. The paper is well-written and easy to follow.

3. The technical part of this paper seems sound to me.

4. Theoretical analyses such as memory capacity round up good work.


**Weaknesses:**

1. Besides Hopfield-like associative memory (AM), there is another class of AM namely predictive coding network (PCN)-based memory. Some recent works including [1,2,3] have shown superior performance of PCN-based approaches compared with Hopfield-like baselines. Seems [3] also considers associative memories for storing and retrieving sequential manner input. I am interested in both theoretical and empirical comparisons between the proposed method and the PCN-like methods.

2. Hopfield network is susceptible to convergence to local minima during the pattern retrieval process. Local minima are stable states that are not the desired patterns. Just wondering, if there are any theoretical guarantees/analyses of convergence of the proposed method.

3. Scaling up Hopfield networks to handle large-scale problems can be challenging. As the number of neurons increases, the computational and memory requirements grow rapidly. Is there any remedy for the efficiency concern, and what is the scalability of the proposed method?

4. Can the authors comment on the utility/challenges in applying their proposed method to real-world datasets/tasks beyond the MNIST datasets used in their experiments? e.g., using them in large-scale language modeling tasks where transformers are now popular.

[1] Salvatori, Tommaso, et al. "Associative memories via predictive coding." Advances in Neural Information Processing Systems 34 (2021): 3874-3886.

[2] Yoo, Jinsoo, and Frank Wood. "BayesPCN: A Continually Learnable Predictive Coding Associative Memory." Advances in Neural Information Processing Systems 35 (2022): 29903-29914.

[3] Tang, Mufeng, Helen Barron, and Rafal Bogacz. "Sequential Memory with Temporal Predictive Coding." arXiv preprint arXiv:2305.11982 (2023).


**Questions:**

1. Comparison with PCN-based approaches.

2. Is there any guarantee or analysis of the convergence?

3. Scalability of the proposed method.

4. Performance on real-world large-scale datasets/settings.

My final score will largely depend on the rebuttal and discussion with the other reviewers.

**Limitations:**

No negative societal impact was found in this paper. I have put my concerns in the "Weaknesses" section.

---

> ### Author Rebuttal · Authors · 2023-08-09
>
> Thank you for your insightful comments and suggestions. We have gone through the weaknesses and addressed them point by point. We are happy to go into more detail for any of these responses.
>
> > Besides Hopfield-like associative memory (AM), there is another class of AM namely predictive coding network (PCN)-based memory. Some recent works including [1,2,3] have shown superior performance of PCN-based approaches compared with Hopfield-like baselines. Seems [3] also considers associative memories for storing and retrieving sequential manner input. I am interested in both theoretical and empirical comparisons between the proposed method and the PCN-like methods.
>
> We have seen the work on the Predictive Coding Network as an alternative mechanism of for associative memory, and found that it to be very interesting. [3] does indeed consider a similar model and is contemporarenous with our submission, as it was posted to the arXiv on May 19, two days after the NeurIPS paper submission deadline. However, given its relationship to our model, we are ready to cite it in the camera-ready work. Our paper is primarily focused on extending Modern Hopfield Networks, deriving theoretical bounds on capacity and verifying them with numerical simulation. [3] introduces a related model, and even mentions that temporal predictive coding can be viewed as a "classical Asymmetric Hopfield Network (AHN) with an implicit statistical whitening process, which leads to more stable performance in sequential memory tasks of structured inputs." We would be interested in exploring theoretical comparisons between the proposed models and the PCN in future work. [1] provides a brief comparison for the symmetric setting, which we anticipate will extend to the sequence setting as well:
>
> - "\[Modern Hopfield Networks\] are able to exactly retrieve data points, while \[the Predictive Coding Network\] always presents a tiny amount of error, even if not visible by the human eye. However, the retrieval process of our model is significantly better, as it always converges to a plausible solution, even when provided with a tiny amount of information, or a large amount of corruption. For example, our model never converges to wrong data points: when tested on complex tasks, it simply outputs fuzzier memories, instead of perfect but wrong reconstructions, as the MHNs."
>
> > Hopfield network is susceptible to convergence to local minima during the pattern retrieval process. Local minima are stable states that are not the desired patterns. Just wondering, if there are any theoretical guarantees/analyses of convergence of the proposed method.
>
> The Hopfield network will converge to these local minima, which we refer to in the paper as "metastable states," if there is too much crosstalk when computing the update. This crosstalk is diminished by adding the nonlinear interaction term, and increasing the nonlinearity generally increases the probability of appropriate recall. The theoretical calculations done in the appendix focus around bounding this crosstalk. There is also previous work (Krotov, Hopfield 2016; Ramsauer et al. 2020; Demircigil et al. 2017) in symmetric modern Hopfield Network which provide theoretical guarantees of convergence, and these calculations will likely carry over to the asymmetric case, at least for the polynomial DenseNet.
>
> > Scaling up Hopfield networks to handle large-scale problems can be challenging. As the number of neurons increases, the computational and memory requirements grow rapidly. Is there any remedy for the efficiency concern, and what is the scalability of the proposed method?
>
> The proposed method massively increases the capacity and computational efficiency of the Hopfield Network without increasing the network size. This is easily implemented since it simply requires applying a polynomial or exponential nonlinearity. However, Hopfield Networks along with any other associative memory method suffer from the fact that in order to retrieve patterns, then those patterns have to be stored somewhere. In the case of the DenseNet, there are no weights that would need to be stored, only the patterns. Furthermore, the distinguishing factor of the DenseNet is its ability to provide error-correction capabilities for robust recall of extremely long sequences, and can be easily implemented for real-time control applications.
>
> > Can the authors comment on the utility/challenges in applying their proposed method to real-world datasets/tasks beyond the MNIST datasets used in their experiments? e.g., using them in large-scale language modeling tasks where transformers are now popular.
>
> In the global rebuttal, we include simulations on the Moving MNIST dataset. The model currently proposed and analyzed has been formulated for discrete-valued patterns, but can be extended to store continuous-valued patterns as required for the transformer architecture. We believe that the model holds potential which we will investigate this in future work, but the focus of the current paper is to introduce the model, derive theoretical capacity limits, and verify it with empirical simulations. In particular, there has been an interest in utilizing state space models as an alternative to transformers for long sequence modeling, which we are exploring for potential connections.
> - Albert Gu, Karan Goel, Christopher Ré 2022
> - Poli et al. 2023
> - Orvieto et al. 2023

---

> ### Comment · Reviewer_mTmK · 2023-08-14
> **Response to Authors**
>
> I appreciate the authors' response and the newly added experiments, which look promising to me. Based on that, I have increased my score to 6.

---

### Author Rebuttal · Authors · 2023-08-10

Thank you for the insightful comments and suggestions. We found that there were some common themes across the reviewers' comments: sequence retrieval for correlated patterns, robust recall of patterns under noise, and a comparison with other sequence recall methods. We individually respond to each reviewer's detailed comments point-by-point.

To address the problem of correlated patterns, Reviewer 2PbW suggested we test our model on the Moving MNIST dataset, which we were not previously aware of and are grateful for the suggestion. The dataset is constructed so that there are $10,000$ sequences of $20$ images, in which two handwritten digits from the MNIST datasets are slowly moving around. This leads to significant amounts of overlap between images within a sequence and should be a great test for robust recall of correlated patterns.

In Figure 1, we demonstrate the effect of pattern overlap on sequence recall and the impact of nonlinear interaction functions on overcoming this effect. In the top row, we show a portion of the ground truth sequence of digits $0$ and $3$ moving around. Despite using a network of size $N = 64 * 64 = 4096$, the linear SeqNet is unable to successfully recall the sequence. For the Polynomial DenseNet, increasing the polynomial degree $d$ slowly increases recall until $d=15$ when there is finally perfect recall. The Exponential DenseNet perfectly recalls the pattern.

In Figure 2, we test the limits of our model by taking all $10,000$ sequences of $20$ Moving MNIST images and concatenating them into a single sequence $200,000$ of images. Here, we display a portion of the ground truth sequence taken from the center, starting in the middle of one subsequence of $2$ and $5$ and ending in the middle of another subsequence of $7$ and $4$. The SeqNet and Polynomial DenseNet are not able to recall anything at all until $d=50$, and the Polynomial DenseNet does not perfectly recall the sequence until $d=70$. The Exponential DenseNet again perfectly recalls the entire sequence.

To address the problem of robust recall, Reviewer 2PbW suggested we test our model by perturbing the initial pattern in the sequence and Reviewer 6yTi suggested we generate new sequences with noise in each pattern in the sequence. In Figure 3, we tried to combine both of these into a single simulation where we simulated a 20 image sequence of $4$ and $9$ and added a randomly flipped bits at each transition. Note that a bitflip probability of 0.5 corresponds to complete noise as a bitflip probability of 1.0 will simply invert the picture. In the top row, we show the input into the network where the probability of pixel being flipped at random is 0.2. The second row shows target pattern the network should transition to. The next few rows show SeqNet and the Polynomial DenseNet failing to recall the sequence until $d=20$. Finally, the last row shows a modified form of the Exponential DenseNet with a smaller base, replacing the nonlinear interaction function $f(x) = e^{(N-1)(x-1)}$ with $f(x) = b^{(N-1)(x-1)}$ where $b = 1.01$. This is because the network size $N = 64*64 = 4096$ results in extremely large negative values in the exponent therefore floating point overflow. Reducing the base preserves the desired robust recall while circumventing numerical limitations.

Finally, we provide a brief survey of related models:
- The Predictive Coding Network (Tang, Barron, Bogacz 2023) is a contemporaneous model which the authors describe as a "classical Asymmetric Hopfield Network (AHN) with an implicit statistical whitening process, which leads to more stable performance in sequential memory tasks of structured inputs." While this model always results in some error, it generally converges to a plausible solution and avoids metastable states. On the other hand, our model provides perfect recall when it works but for insufficient nonlinearity converges to metastable states.
- Whittington et al. 2022 introduces a connection between the Transformer architecture and the Tolman-Eichenbaum machine, a model of the hippocampus, and propose the usage of an external network to learn a positional encoding via path integration. We propose a model that does not require an external network, although we briefly explore the possibility of adding an external network for context-dependent gating in the context of motor sequence control in lines 314-317.
- Karuvally et al. 2022 extends Modern Hopfield Networks to the sequence setting by developing a way to analyze an energy function in terms of an adiabatic limit. The GSEMM is related to the biologically-plausible implementation of our Mixed Network. Furthermore, they analyze the model in the setting of continuous-time dynamics, allowing for different timescales between the hidden and feature layers, and also allow for intralayer synapses within the hidden layer. For theoretical tractability, we focus on the discrete-time setting in order to assess the sequence capacity of the model, but our model can be easily extended to continuous time and continuous patterns.
- Muscinelli et al 2017 propose a way to construct a sequence of $2^N$ patterns an update rule to perfect recall this sequence. but this model is unable to store arbitrary patterns and lacks the error-correction capabilities that our model possesses.
- The following works focus on overcoming capacity limitations of correlated patterns: Diederich, Opper. PRL, 1987 proposes 2 local learning rule to store correlated patterns where each pattern must be learned sequentially. Gardener 1987 and Bressloff, Taylor 1992 demonstrates ways to store temporal sequences by utilizing perceptron-style learning rules. Blumenfeld et al. 2006 stores correlated patterns by changing weights proportionally to the difference between input and stored memories. These models all overcome pattern correlation via an iterative learning rule which explicitly accounts for correlation, whereas our model can store patterns without doing so.

---

### Decision · Program_Chairs · 2023-09-21

**Decision:**

Accept (poster)

**Comment:**

Meta Review for Long Sequence Hopfield Memory

As summarized by Reviewer 6yTi, this work looks at combining two modifications to the basic Hopfield network to create a network capable of creating and recalling long sequential memories (i.e. sequences of states of the network). The sequence capacity of the proposed network is bounded analytically, and supported by numerical simulations. This model is then extended to allow memories to be recalled a fixed number of times before transitioning to the next item in the sequence. Finally, an implementation of this network using more biologically plausible neurons is given.

Most authors agree on the novelty of the work which extends the original Hopfield network, that the paper is well written, and the theory and simulations complement each other well. A clear acceptance.